# Invariant Representation Learning for Source-Free Time Series Forecasting with LLM-Centric Proxy Denoising

**Kangjia Yan** [* 1]  **Chenxi Liu** [* 2]  **Hao Miao** [† 3]  **Xinle Wu** [4]  **Yan Zhao** [5]  **Chenjuan Guo** [1]  **Bin Yang** [† 1]

## Abstract

Effective time series forecasting enables various real-world applications, benefiting from the proliferation of mobile devices. However, the volume of time series data may vary significantly across domains due to high data acquisition costs and data regulations. To maximally create value from sparse data, this study focuses on a new problem of source-free time series forecasting, aiming to adapt a pretrained model from sufficient source time series to the sparse target time series without access to the source data, enabling data protection. To achieve this, we propose TimeID, a novel source-free time series forecasting framework with a large language model (LLM) centric proxy denoising inspired by the powerful generalization capabilities of LLMs. Specifically, TimeID consists of three key components: (1) dual-branch invariant disentangled feature learning that enforces representation- and gradient-wise invariance by means of season-trend decomposition; (2) lightweight, parameter-free proxy denoising that dynamically calibrates systematic biases of LLMs; and (3) knowledge distillation that bidirectionally aligns the denoised prediction and the original target prediction. Extensive experiments on real-world datasets demonstrate that TimeID outperforms state-of-the-art baselines, improving MSE and MAE by 10.7% and 9.3% on average. The code is available at https://github.com/decisionintelligence/TimeID.

---

[*]Equal contribution [†]Corresponding authors [1]School of Data Science and Engineering, East China Normal University, Shanghai, China [2]CAIR, Hong Kong Institute of Science and Innovation, Chinese Academy of Sciences, Hong Kong, China [3]Department of Computing, Hong Kong Polytechnic University, Hong Kong, China [4]National University of Singapore, Singapore [5]Shenzhen Institute for Advanced Study, University of Electronic Science and Technology of China, Shenzhen, Guangdong, China. Correspondence to: Bin Yang <byang@dase.ecnu.edu.cn>, Hao Miao <hao.miao@polyu.edu.hk>.

*Proceedings of the $43^{rd}$ International Conference on Machine Learning*, Seoul, South Korea. PMLR 306, 2026. Copyright 2026 by the author(s).

## 1. Introduction

The widespread deployment of Internet-of-Things (IoT) sensors has produced massive time series data across domains (Sun et al., 2025; Wang et al., 2024a; Wei et al., 2026), including traffic (Kieu et al., 2024; Cirstea et al., 2022b; Chen et al., 2026; Lu et al., 2011; Ma et al., 2014), weather (Hettige et al., 2024; Tian et al., 2026), and energy (Wu et al., 2020). Accurate time series forecasting is crucial, enabling effective decision-making across diverse domains (Liu et al., 2025b; 2024a; Chen et al., 2023; Pan et al., 2023; Cheng et al., 2023). We are seeing impressive advances in machine learning, especially in deep learning, that are successful in effective feature extraction and value creation (Hettige et al., 2024; Liu et al., 2025c). They are mainly dedicated to creating models based on large amounts of domain-specific time series (see Figure 1(a)). However, time series data can be sparse for various reasons, such as high data acquisition costs, data regulations and data privacy. The performance of existing time series forecasting methods may degrade remarkably with such insufficient training data (Jin et al., 2022).

Although recent research efforts have been devoted to addressing sparse training data by means of transfer learning (Shi et al., 2023; Wang et al., 2024a), these are mainly designed for computer vision and natural language processing while ignoring the specific characteristics of time series, i.e., capturing complex temporal correlations (Shao et al., 2025; Yang et al., 2023). In addition, these methods are often performed across domains by leveraging both source and target data (Liu et al., 2024b). However, the reliance on source data may raise various concerns, e.g., training inefficiency and data privacy. Further, large language models (LLM) based methods emerge as a new paradigm for universal time series forecasting (Liu et al., 2025a; Xu et al., 2026). Nonetheless, it is expensive to train these large models, incurring high computational costs. Further, despite LLM-based methods offering acceptable time series forecasting performance, they often fail to achieve superior performance on specific domains, especially domains with scarce time series. To address these issues, this study focuses on a new problem of source-free domain adaptation (Tang et al., 2025; Ragab et al., 2023; Campos et al.,

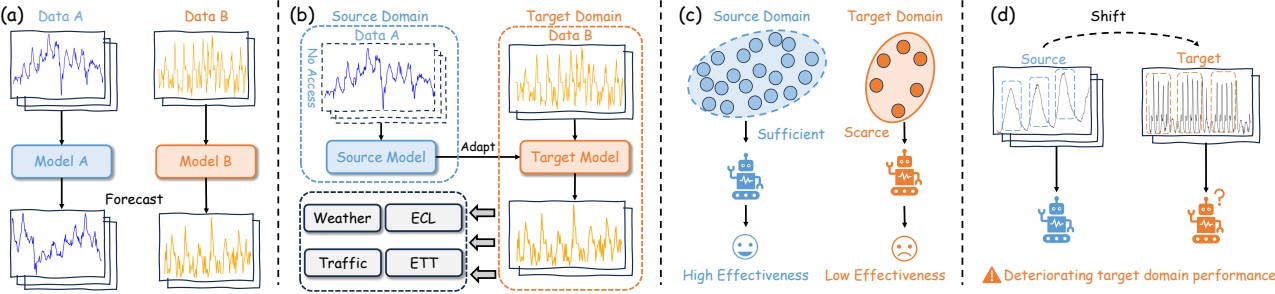

*Figure 1.* (a) Domain-specific time series forecasting. (b) Source-free time series forecasting. (c) Limited target data acquisition. (d) Cross-domain distribution shift.

2024) for time series forecasting, referred to as source-free time series forecasting (SF-TSF), as shown in Figure 1 (b). The SF-TSF aims to directly adapt a source model to a target domain using only its parameters, without accessing its source data.

However, it is non-trivial to develop SF-TSF methods due to the following challenges. First, in the source-free domain adaptation scenarios, it is often hard to acquire sufficient target data due to privacy and data collection mechanisms. Limited target data acquisition often results in insufficient observations, as shown in Figure 1 (c). It is challenging to effectively capture the complex temporal correlations of such sparse target data. Second, alleviating the cross-domain distribution shift between source and target data also poses difficulties. Owing to differences in data sensing mechanisms, the statistical properties of time series, such as trend and season, of the target domain may largely deviate from those of the source domain. The distribution shift makes it difficult for most existing time series modeling methods trained on the source domain to generalize to the target domain, as shown in Figure 1 (d), resulting in low effectiveness. Recent advances have incorporated LLMs into time series modeling, benefiting from their pre-trained knowledge and generalization ability (Huang et al., 2024). However, this naturally introduces the third challenge: how to effectively leverage the knowledge embedded in LLMs while alleviating noise, since LLMs are prone to hallucinations (Sriramanan et al., 2024), producing irrelevant or misleading outputs when faced with domain-scarce signals, which could distort the forecasting.

This study addresses the above challenges by providing a novel Source-Free **Time** Series Forecasting framework with **I**nvariant feature disentanglement and proxy **D**enoising (TimeID). To facilitate effective temporal correlation extraction across sparse target time series, we develop an innovative source-free domain adaptation paradigm, where the target model borrows the rich knowledge learned on sufficient source data based on the assumption that time series from different domains share certain latent patterns (Jin et al., 2022). Further, we achieve completely invariant disentangled feature learning, which also alleviates cross-domain

distribution shift. Specifically, we design a dual-branch architecture that explicitly decomposes input series into seasonal and trend components and enforces invariance at both the representation and gradient levels. Stochastic augmentation and specialized invariance blocks further strip away component-specific cues, obtaining disentangled and component-invariant representations. To leverage the transferability ability of LLMs while alleviating hallucinations, we apply pre-trained LLMs to guide the target model, alleviating the impact of domain shift on the target model. Then, we introduce a proxy denoising mechanism, which treats LLM as a powerful but probably noisy proxy forecaster, to denoise the LLM's forecasts. It dynamically corrects its systematic bias on the target domain by leveraging the consensus between the source model and the adapting target model, producing more reliable forecasts for subsequent guidance. Then, we establish a bidirectional knowledge transfer loop: denoised proxy forecasts supervise the target model, while target predictions feed back to stabilize the proxy correction, preventing distribution drift from the target domain. Finally, we employ knowledge distillation to further calibrate the target prediction with the denoised prediction, enhancing model performance.

The main contributions are summarized as follows:

- To the best of our knowledge, this is the first study to learn source-free time series forecasting and propose an LLM-empowered framework called TimeID that unleashes the power of LLMs and models trained on sufficient data to improve.

- We propose an invariant disentangled feature learning method to handle the cross-domain distribution shift, a proxy denoising strategy to alleviate the hallucinations of LLMs, and a knowledge distillation mechanism to transfer denoised knowledge from LLMs to a lightweight target model.

- Extensive experiments on real-world datasets demonstrate that TimeID outperforms state-of-the-art baselines, achieving average improvements of **10.7%** and **9.3%** in terms of MSE and MAE, respectively, offering a brand new paradigm for cross-domain time series analytics.

## 2. Related Work

**Time Series Forecasting.** Time series forecasting attracts increasing interest due to the growing availability of time series data and rich downstream applications (Benidis et al., 2022; Hettige et al., 2024; Wu et al., 2024). Traditional time series forecasting models (Box et al., 2015) are mostly based on shallow statistics, making them difficult to capture the complex temporal correlations. Recent advances in deep learning techniques apply neural networks for effective time series modeling (Zhou et al., 2022; Bai et al., 2020), including various architectures, e.g., CNNs (Wu et al., 2023), RNNs (Lai et al., 2018), and Transformers (Wu et al., 2021; Liu et al., 2024c; Xia et al., 2025; Qiu et al., 2025; Cirstea et al., 2022a). LLM-based methods (Liu et al., 2025c; Zhou et al., 2023; Jin et al., 2024) emerge as a new paradigm for time series forecasting empowered by their powerful general feature extraction capabilities. However, most existing methods require sufficient training data. Their performance may degrade remarkably when training on sparse data.

**Source-free Domain Adaptation (SFDA).** SFDA adapts pre-trained models to target domains without accessing source data (Kundu et al., 2020; Kim et al., 2021; Fang et al., 2024; Mitsuzumi et al., 2024). Different from time series foundation model (Wang et al., 2025b;a; Wu et al., 2026), which is pretrained on multi-domain data, the source model in SFDA is trained on one source dataset. For example, SHOT (Liang et al., 2020) leverages information maximization and self-supervised pseudo-labeling. NRC (Yang et al., 2021) introduces neighborhood clustering to improve adaptation stability. However, these methods are primarily tailored for computer vision and natural language processing (Li et al., 2024), and cannot capture the unique temporal correlations among time series. Although recent studies (Ragab et al., 2023; Zhong et al., 2025; Ragab et al., 2026) have explored SFDA for time series imputation, its application to forecasting remains largely underexplored. At the same time, large language models (LLMs) have demonstrated the ability to acquire generalized knowledge across diverse tasks, showing strong potential for time series forecasting (Jin et al., 2024). However, most existing SFDA approaches fail to harness the knowledge of LLMs effectively.

## 3. Methodology

Figure 2 presents the overview of TimeID, which integrates an invariant feature disentanglement learning module, a proxy denoising module, and a knowledge distillation module. Sequentially, TimeID begins with training a source model $\theta_s$ on source data. Without revisiting the source dataset, $\theta_s$ is copied to initialize the target model $\theta_t$, which adapts to the target domain. Meanwhile, a pre-trained LLM $\theta_{ts}$ is applied for extracting knowledgeable features, which

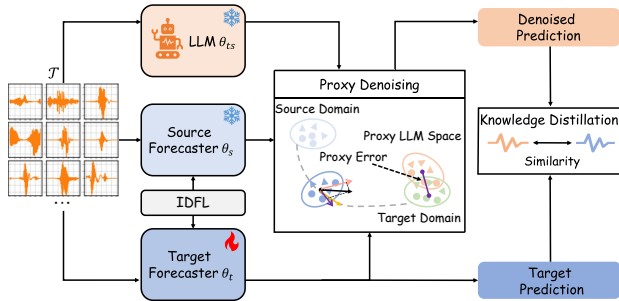

Figure 2. TimeID Framework Overview. Invariant Disentangled Feature Learning (IDFL) is designed to boost forecasters to learn invariant features by disentangling the seasonal and trend components. Proxy Denoising aims to denoise the LLM's outputs.

are further calibrated by the proxy denoising module to alleviate hallucinations. Finally, the knowledge distillation module is designed to minimize disagreement between the corrected proxy forecasts and target predictions.

### 3.1. Invariant Disentangled Feature Learning

Invariant Disentangled Feature Learning (IDFL) aims to handle cross-domain distribution shift. As shown in Figure 3, IDFL consists of a decomposition block, forecasters, a representation-invariant block, a gradient-invariant block, and a Fourier transform module. It decomposes the input series into trend and seasonal components and learns component-invariant representations. The invariant features remain stable while other factors change (Parascandolo et al., 2021). For example, trend features should stay consistent under seasonal variations, and vice versa. Such disentangled invariants enhance forecasting accuracy across domains. We design two complementary branches: the *trend branch*, where seasonal variations act as domains; and the *seasonal branch*, where trend variations act as domains. To improve generalization, invariance is explicitly enforced at both the representation level and the gradient level. Finally, the IDFL yields disentangled seasonal-trend features that provide accurate prediction.

**Decomposition Module.** To begin with, the time series is decomposed into seasonal and trend features. Given a time series with $C$ features $\mathcal{T} \in \mathbb{R}^{B \times L \times C}$, we extract the trend component by a moving average kernel of length $k_{trend}$:

$$\mathbf{t} = AvgPool_{k_{trend}}(\mathcal{T}), \mathbf{s} = \mathcal{T} - \mathbf{t}, \qquad (1)$$

where $\mathbf{t} \in \mathbb{R}^{B \times L \times C}$ and $\mathbf{s} \in \mathbb{R}^{B \times L \times C}$ denote the trend and seasonal signals, respectively.

To diversify the decomposed trend and seasonal components, we perform stochastic forward passes by applying dropout in the decomposition module and input the same time series $\mathcal{T}$ into the decomposition module twice, i.e., $\mathbf{s}^{(1)}, \mathbf{t}^{(1)} = Decomposition(\mathcal{T})$ and $\mathbf{s}^{(2)}, \mathbf{t}^{(2)} = Decomposition(\mathcal{T})$. Next, features $(\mathbf{s}^{(1)}, \mathbf{s}^{(2)})$ and $(\mathbf{t}^{(1)}, \mathbf{t}^{(2)})$ are separately fed into the forecasters.

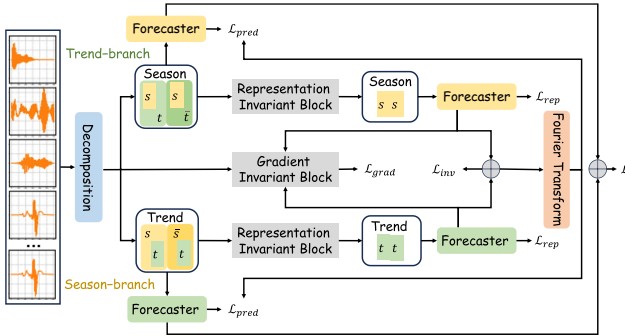

*Figure 3.* Model Training via **I**nvariant **D**isentangled **F**eature **L**earning (IDFL).

**Forecaster.** The forecaster is implemented as a lightweight Time-Series Feature Extractor (TSFE) (Miao et al., 2024), consisting of patching with patch length $P$ and stride $S$, $N_{op}$ stacked self-attention and feed-forward networks, and a linear layer that outputs latent features:

$$\mathbf{z}_{tre}^{(i)} = Forecaster_{tre}(\mathbf{t}^{(i)}), \ \mathbf{z}_{sea}^{(i)} = Forecaster_{sea}(\mathbf{s}^{(i)}), \quad (2)$$

where $i \in \{1, 2\}$ and $\mathbf{z}_{tre}, \mathbf{z}_{sea} \in \mathbb{R}^{B \times N \times C}$ are the prediction of trend and seasonal features. Given the mean squared error loss function $l(\cdot, \cdot)$, the time series forecasting loss can be defined as:

$$\mathcal{L} = l(\mathbf{z}_{tre} + \mathbf{z}_{sea}, y), \quad (3)$$

where $y$ represents the ground truth for the input time series.

**Representation-Level Alignment.** Representation-level alignment is designed to force samples that share a pattern, no matter which domain they come from, to occupy the same region of the feature space. Concretely, the network learns a single mapping that pushes every domain's distribution toward one common statistical form. Here, we denote the forecasting task set as $\mathcal{C} = \{seasonal, trend\}$. We use $\mathbf{s}, \bar{\mathbf{s}}$ to represent different seasons, and use $\mathbf{t}, \bar{\mathbf{t}}$ to represent different trends so that we can denote decomposed features as $\mathcal{F} = \{(\mathbf{s}, \mathbf{t}), (\mathbf{s}, \bar{\mathbf{t}}), (\mathbf{t}, \mathbf{s}), (\mathbf{t}, \bar{\mathbf{s}})\}$, where $\{(\mathbf{s}, \mathbf{t}), (\mathbf{s}, \bar{\mathbf{t}})\}$ represent $\mathbf{s}^{(1)}, \mathbf{s}^{(2)}$, $\{(\mathbf{t}, \mathbf{s}), (\mathbf{t}, \bar{\mathbf{s}})\}$ represent $\mathbf{t}^{(1)}, \mathbf{t}^{(2)}$. For example, $(\mathbf{s}, \mathbf{t})$ represents a seasonal feature with a different trend than another seasonal feature $(\mathbf{s}, \bar{\mathbf{t}})$, considering that seasonal and trend features have not yet been fully disentangled.

Initially, we derive the gradient of the model on each branch with respect to the representations:

$$g_j^i = \frac{\partial(Forecaster_j(Decomposition(\mathcal{T}))}{\partial \boldsymbol{\mathcal{X}}}, \quad (4)$$

where $i \in \mathcal{F}, j \in \mathcal{C}$. $\boldsymbol{\mathcal{X}}$ denotes the embedding of input time series $\mathcal{T}$. The representations associated with similar gradients indicate intrinsic characteristics of seasonal

patterns that are invariant to trend factors or vice versa. Consequently, we compute the absolute value of the difference between the two gradients:

$$\Delta g_{sea} = |g_{sea}^{(\mathbf{s},\mathbf{t})} - g_{sea}^{(\mathbf{s},\bar{\mathbf{t}})}|, \Delta g_{tre} = |g_{tre}^{(\mathbf{t},\mathbf{s})} - g_{tre}^{(\mathbf{t},\bar{\mathbf{s}})}|. \quad (5)$$

The variables with a small difference correspond to seasonal features that are insensitive to trend variation and trend features that are insensitive to seasonal variation. We rank the absolute gradient differences in descending order and then take the $\alpha$-percentile value, denoted as $d^\alpha$. A binary mask $m$ of identical shape to the representation is then generated. For the $k$-th element,

$$m_j(k) = \begin{cases} 0, & \Delta g_j(k) \geq d^\alpha \\ 1, & else \end{cases}. \quad (6)$$

By applying the mask to the original representation, the network filters out component-varying feature variables to learn the invariant seasonal feature $\hat{\mathbf{s}}$ and invariant trend feature $\hat{\mathbf{t}}$, i.e., $\hat{\mathbf{s}} = \boldsymbol{\mathcal{X}} \odot m_{sea}, \hat{\mathbf{t}} = \boldsymbol{\mathcal{X}} \odot m_{tre}$. Then, the learned invariant features are fed into the forecaster:

$$\hat{\mathbf{z}}_{tre} = Forecaster_{tre}(\hat{\mathbf{t}}), \hat{\mathbf{z}}_{sea} = Forecaster_{sea}(\hat{\mathbf{s}}). \quad (7)$$

Finally, the mean squared error is defined as the loss of invariant features in predictions,

$$\mathcal{L}_{inv} = l(\hat{\mathbf{z}}_{tre} + \hat{\mathbf{z}}_{sea}, y), \quad (8)$$

where $y$ represents the ground truth of the input time series.

**Fourier Transform Module.** Average pooling provides a coarse time-domain decomposition into trend and residual, while Fourier transform offers a fine-grained frequency-domain analysis to capture periodic components. This module aims to provide frequency-consistent supervision for invariant learning. Time-series windows are decomposed in the time domain via the Discrete Fourier Transform (DFT). Given the whole time series embedding $\boldsymbol{\mathcal{X}} \in \mathbb{R}^{B \times L \times E}$, we treat each channel independently. The DFT of each channel is defined as:

$$\mathbf{X}[k] = \sum_{t=0}^{L-1} \mathbf{X}[t] \exp\left(-\frac{2\pi i}{L}kt\right), \quad k = 0, \ldots, L-1. \quad (9)$$

Specifically, frequency coefficients are split into low-frequency (trend) and high-frequency (seasonality) subsets using a predefined cut-off index $k_{cut}$, which is determined by the length of the frequency spectrum. We set the seasonal ratio to 20%:

$$\mathbf{X}_{\text{tr}}[k] = \begin{cases} \mathbf{X}[k], & 0 \leq k \leq k_{cut}, \\ 0, & otherwise, \end{cases}$$

$$\mathbf{X}_{sea}[k] = \begin{cases} \mathbf{X}[k], & k_{cut} < k \leq \lfloor L/2 \rfloor, \\ 0, & otherwise. \end{cases} \quad (10)$$

Then, the inverse DFT $\mathcal{F}^{-1}(\cdot)$ is applied to obtain the trend $\mathbf{t}$ and seasonal signals $\mathbf{s}$, respectively: $\mathbf{t} = \mathcal{F}^{-1}(\mathbf{X}_{tr})$, $\mathbf{s} = \mathcal{F}^{-1}(\mathbf{X}_{sea})$. After the representation-level alignment, we feed the predictions of invariant features into Fourier transform module $FT(\cdot)$, getting the decomposed seasonal and trend features:

$$\mathbf{s}', \mathbf{t}' = FT(\hat{\mathbf{z}}_{tre} + \hat{\mathbf{z}}_{sea}). \tag{11}$$

Finally, we compute the loss function with the new decomposed features and the prediction of raw decomposed features:

$$\begin{aligned}
\mathcal{L}_{pred} = &\sum_{i=1}^{2} l(Forecaster_{sea}(\mathbf{s}^{(i)}), \mathbf{s}') \\
&+ \sum_{i=1}^{2} l(Forecaster_{tre}(\mathbf{t}^{(i)}), \mathbf{t}').
\end{aligned} \tag{12}$$

The loss function $\mathcal{L}_{rep}$ is the combination of the trend-irrelevant seasonal-specific representation and the seasonal-irrelevant trend-specific representation:

$$\mathcal{L}_{rep} = l(Forecaster_{tre}(\hat{\mathbf{t}}), \mathbf{t}') + l(Forecaster_{sea}(\hat{\mathbf{s}}), \mathbf{s}'). \tag{13}$$

**Gradient-Level Alignment.** Gradient-level alignment aims to optimize the trajectories of all branches toward a common direction. By explicitly shrinking the dispersion of inter-branch gradients, the model is encouraged to discard component-specific cues and retain invariant ones. We derive the gradient of seasonal predictions with respect to the seasonal forecaster under varying trends, that of trend predictions under varying seasonal components, as detailed below:

$$\begin{aligned}
G_{sea}^i &= \frac{\partial l(Forecaster_{sea}(\mathbf{s}^i), \mathbf{s}')}{\partial \theta_{sea}}, \\
G_{tre}^i &= \frac{\partial l(Forecaster_{tre}(\mathbf{t}^i), \mathbf{t}')}{\partial \theta_{tre}},
\end{aligned} \tag{14}$$

where $\theta$ denotes the parameters of $Forecaster(\cdot)$ and decomposition module $Decomposition(\cdot)$.

Steering every branch along this identical route markedly eases the acquisition of invariant predictions (Chen et al., 2025b). To enforce this gradient-level alignment and distill disentangled invariances, we suppress the model's ability to identify patterns by minimizing the Euclidean distance, denoted $d_{euc}(\cdot, \cdot)$, between the respective gradient vectors, formulated as:

$$\mathcal{L}_{grad} = d_{euc}(G_{sea}^{(s,t)}, G_{sea}^{(s,\bar{t})}) + d_{euc}(G_{tre}^{(t,s)}, G_{tre}^{(t,\bar{s})}). \tag{15}$$

Therefore, the gradient-level alignment drives all parameter updates along a unified trajectory, thereby strengthening the robustness of the forecaster.

### 3.2. Proxy Denoising

LLMs for time series modeling benefit from their pre-trained knowledge and generation ability, but would introduce hallucinations. The proxy denoising (PD) is proposed to quantify the proxy error of the LLMs and then generate calibrated predictions for enhancing prediction. LLM prediction errors relate to domain differences, which correlate with the forecasting biases due to the domain difference between source and target datasets. So we leverage the disagreement between the source model $\theta_s$ and the target model $\theta_t$ to estimate and suppress the noise dynamically. $\theta_s$ encodes knowledge acquired on the source distribution, remaining oblivious to target-specific drift, while $\theta_t$ is trainable and gradually adapts to the target. Its current state reflects the best in-domain hypothesis available at any moment.

When all three models agree, the LLM is likely reliable. If $\theta_s$ and $\theta_t$ agree with each other but deviate from the LLM, the discrepancy is interpreted as proxy noise that needs to be corrected. For every target mini-batch $B_t = \{x_i\}_{i=1}^{B}$, we compute the prediction of three models: $z_{ts,i} = \theta_{ts}(x_i)$, $z_{s,i} = \theta_s(x_i)$, $z_{t,i} = \theta_t(x_i)$. The per-sample noise vector is simply the signed residual $e_i = \theta_s(x_i) - \theta_t(x_i)$, which captures how far the LLM predictions deviate from the consensus of source and target models. The subtraction serves as an empirical error signal. The estimated noise is subtracted from the LLM outputs to obtain the denoised prediction:

$$\tilde{z}_i = \theta_{ts}(x_i) - \alpha(\theta_s(x_i) - \theta_t(x_i)), \tag{16}$$

where $\theta_s, \theta_t, \theta_{ts}$ apply the source model, target model, and LLM to get the corresponding prediction, and $\alpha$ represents the correction strength, which is a hyperparameter. Particularly, $\alpha = 1$ performs full correction (complete trust in the source-target consensus) and $\alpha = 0$ retains the raw LLM predictions. The denoised predictions $\tilde{z}_i$ are forwarded to the subsequent knowledge distillation.

### 3.3. Knowledge Distillation

To improve inference efficiency, LLM's outputs are distilled to a lightweight target model to guide the model with purified knowledge and prevent the LLM from drifting away from the target domain. The output of the target model is aligned with the corrected proxy via Mean Squared Error:

$$\mathcal{L}_{kd} = l(\theta_{ts}(x_i) - \alpha(\theta_s(x_i) - \theta_t(x_i)), \theta_t(x_i)). \tag{17}$$

Minimizing $\mathcal{L}_{kd}$ pulls the target model's predictions toward the denoised large language model without any label supervision. The gradient flow is one-way: only the target model $\theta_t$ is updated; the large language model remains frozen. Consequently, the target model receives high-level temporal knowledge distilled from the denoised LLM while preserving its own low-rank adaptation capacity.

*Table 1.* Overall Performance Comparison.

| Methods | Dataset | ETTh1 → ETTh2 | | | ETTh1 → ETTm1 | | | ETTh1 → ETTm2 | | | ETTh1 → Weather | | | ETTh1 → Electricity | | | ETTh1 → Traffic | | |
|---|---|---|---|---|---|---|---|---|---|---|---|---|---|---|---|---|---|---|---|---|
| | PL | 96 | 192 | 336 | 96 | 192 | 336 | 96 | 192 | 336 | 96 | 192 | 336 | 96 | 192 | 336 | 96 | 192 | 336 |
| DLinear | MSE | 0.287 | 0.367 | 0.438 | **0.357** | 0.396 | 0.428 | 0.180 | 0.240 | 0.301 | 0.178 | 0.220 | **0.262** | 0.178 | 0.191 | 0.217 | 0.460 | 0.484 | 0.522 |
| | MAE | 0.346 | 0.400 | 0.453 | 0.391 | 0.422 | 0.443 | 0.274 | 0.319 | 0.363 | 0.244 | 0.280 | 0.314 | 0.279 | 0.292 | 0.319 | 0.332 | 0.348 | 0.377 |
| TimeKAN | MSE | 0.284 | 0.352 | 0.409 | 0.367 | 0.399 | 0.426 | 0.182 | 0.236 | 0.280 | 0.735 | 0.742 | 0.745 | 1.085 | 1.083 | 1.079 | 0.540 | 0.555 | 0.557 |
| | MAE | 0.343 | 0.387 | 0.429 | 0.399 | 0.417 | 0.431 | 0.267 | 0.303 | **0.329** | 0.665 | 0.666 | 0.666 | 0.853 | 0.852 | 0.852 | 0.828 | 0.831 | 0.829 |
| SimpleTM | MSE | 0.283 | 0.351 | 0.355 | 0.383 | 0.401 | 0.486 | 0.182 | 0.231 | 0.278 | 0.754 | 0.768 | 0.701 | 1.080 | 1.074 | 1.074 | 0.526 | 0.539 | 0.549 |
| | MAE | 0.340 | 0.388 | 0.404 | 0.399 | 0.414 | 0.464 | 0.269 | 0.302 | 0.333 | 0.670 | 0.679 | 0.645 | 0.851 | 0.851 | 0.851 | 0.825 | 0.828 | 0.829 |
| TimesNet | MSE | 0.362 | 0.429 | 0.457 | 0.498 | 0.635 | 0.617 | 0.213 | 0.270 | 0.315 | 0.737 | 0.742 | 0.744 | 1.086 | 1.082 | 1.080 | 0.524 | 0.534 | 0.546 |
| | MAE | 0.408 | 0.437 | 0.461 | 0.462 | 0.534 | 0.526 | 0.297 | 0.331 | 0.358 | 0.665 | 0.665 | 0.668 | 0.853 | 0.852 | 0.852 | 0.825 | 0.827 | 0.828 |
| TimeMixer | MSE | 0.330 | 0.399 | 0.431 | 0.438 | 0.534 | 0.491 | 0.185 | 0.234 | 0.282 | 0.744 | 0.741 | 0.743 | 1.083 | 1.083 | 1.082 | 0.524 | 0.534 | 0.548 |
| | MAE | 0.373 | 0.413 | 0.447 | 0.425 | 0.483 | 0.461 | 0.269 | 0.303 | 0.333 | 0.667 | 0.665 | 0.666 | 0.852 | 0.852 | 0.852 | 0.825 | 0.827 | 0.829 |
| WPMixer | MSE | 0.291 | 0.375 | 0.414 | 0.371 | 0.403 | 0.461 | 0.182 | 0.233 | 0.284 | 0.736 | 0.764 | 0.741 | 1.082 | 1.083 | 1.080 | 0.525 | 0.536 | 0.548 |
| | MAE | 0.348 | 0.399 | 0.432 | 0.392 | **0.409** | 0.441 | 0.270 | 0.302 | 0.337 | 0.663 | 0.677 | 0.665 | 0.852 | 0.852 | 0.852 | 0.825 | 0.827 | 0.829 |
| iTransformer | MSE | 0.358 | 0.489 | 0.510 | 0.402 | 0.420 | 0.561 | 0.194 | 0.237 | 0.312 | 0.189 | 0.237 | 0.274 | 0.180 | 0.191 | 0.219 | 0.452 | 0.479 | 0.516 |
| | MAE | 0.395 | 0.467 | 0.487 | 0.415 | 0.426 | 0.503 | 0.275 | 0.306 | 0.351 | 0.238 | 0.282 | 0.309 | 0.288 | 0.299 | 0.316 | 0.341 | 0.358 | 0.387 |
| FEDformer | MSE | 0.388 | 0.485 | 0.421 | 0.646 | 0.630 | 0.695 | 0.287 | 0.331 | 0.391 | 0.747 | 0.746 | 0.734 | 1.083 | 1.082 | 1.079 | 0.529 | 0.535 | 0.548 |
| | MAE | 0.431 | 0.500 | 0.462 | 0.534 | 0.544 | 0.566 | 0.359 | 0.388 | 0.423 | 0.667 | 0.668 | 0.661 | 0.852 | 0.852 | 0.851 | 0.827 | 0.827 | 0.829 |
| PatchTST | MSE | 0.280 | 0.359 | 0.354 | 0.364 | 0.400 | 0.430 | 0.180 | 0.234 | 0.286 | 0.232 | 0.279 | 0.335 | 0.181 | 0.197 | 0.216 | 0.455 | 0.484 | 0.519 |
| | MAE | 0.340 | 0.388 | 0.399 | 0.398 | 0.420 | 0.438 | 0.267 | 0.303 | 0.338 | 0.282 | 0.318 | 0.357 | 0.278 | 0.292 | 0.313 | 0.333 | 0.345 | 0.382 |
| OFA | MSE | 0.296 | 0.374 | 0.394 | 0.382 | 0.404 | 0.430 | 0.185 | 0.231 | 0.287 | 0.195 | 0.232 | 0.269 | 0.173 | 0.206 | 0.220 | 0.459 | 0.483 | 0.515 |
| | MAE | 0.352 | 0.404 | 0.424 | 0.399 | 0.413 | 0.433 | 0.270 | 0.305 | 0.338 | 0.248 | 0.280 | 0.304 | 0.283 | 0.313 | 0.320 | 0.347 | 0.356 | 0.380 |
| Time-LLM | MSE | 0.324 | 0.374 | 0.393 | 0.402 | 0.424 | 0.456 | 0.183 | 0.236 | 0.283 | 0.176 | 0.224 | 0.272 | 0.171 | 0.192 | 0.220 | 0.452 | 0.478 | 0.513 |
| | MAE | 0.367 | 0.400 | 0.421 | 0.410 | 0.423 | 0.434 | 0.272 | 0.307 | 0.331 | 0.229 | 0.275 | 0.304 | 0.284 | 0.291 | 0.318 | 0.338 | 0.347 | 0.381 |
| TimeID | MSE | **0.280** | **0.345** | **0.346** | 0.359 | 0.392 | 0.422 | **0.177** | **0.230** | **0.277** | **0.169** | **0.219** | 0.265 | **0.170** | **0.187** | **0.211** | 0.452 | **0.474** | **0.510** |
| | MAE | **0.338** | **0.385** | **0.398** | 0.390 | 0.413 | 0.430 | **0.267** | **0.297** | 0.330 | **0.228** | **0.275** | **0.303** | **0.276** | **0.289** | **0.312** | 0.327 | 0.342 | 0.373 |

## 3.4. Overall Objective Function

The final loss consists of a time series forecasting loss $\mathcal{L}$, an invariant features forecasting loss $\mathcal{L}_{inv}$, a disentangled features forecasting loss $\mathcal{L}_{pred}$, a representation invariant loss $\mathcal{L}_{rep}$, a gradient invariant loss $\mathcal{L}_{grad}$ and a knowledge distillation loss $\mathcal{L}_{kd}$. The overall loss is:

$$
\mathcal{L}_{all} = \mathcal{L} + \lambda_{inv}\mathcal{L}_{inv} + \lambda_{pred}\mathcal{L}_{pred} \\
+ \lambda_{rep}\mathcal{L}_{rep} + \lambda_{grad}\mathcal{L}_{grad} + \lambda_{kd}\mathcal{L}_{kd}, \quad (18)
$$

where $\lambda_{inv}, \lambda_{pred}, \lambda_{rep}, \lambda_{grad}, \lambda_{kd}$ are trade-off parameters.

## 4. Experiments

### 4.1. Experimental Setup

#### 4.1.1. DATASETS.

The experiments are carried out on seven widely-used time series datasets, including ETTh1, ETTh2, ETTm1, ETTm2, Weather, Traffic, and Electricity (Liu et al., 2024b).

- **Weather.** The Weather dataset contains 21 indicators of weather(e.g., air temperature and humidity), which are collected in Germany. The data is recorded every 10 minutes.

- **Traffic.** The Traffic dataset contains hourly road occupancy rates obtained from sensors located on San Francisco freeways from 2015 to 2016.

- **Electricity.** The Electricity dataset contains the hourly electricity consumption of 321 clients from 2012 to 2014.

- **ETT.** The ETT dataset includes two hourly-level datasets(ETTh1 and ETTh2) and two 15-minute-level datasets (ETTm1 and ETTm2). Each dataset includes 7 oil and load features of electricity transformers between July 2016 and July 2018.

#### 4.1.2. BASELINES.

We compare TimeID with the following existing methods for time series forecasting.

- **OFA:** Introduces a frozen pretrained Transformer framework that reuses frozen self-attention and feedforward blocks from large pretrained language or vision models, fine-tuning only lightweight adapters to achieve state-of-the-art performance across diverse time series tasks such as forecasting, classification, and anomaly detection (Zhou et al., 2023).

- **SimpleTM:** Introduces a simple yet effective architecture that uniquely integrates classical signal processing ideas with a slightly modified attention mechanism (Chen et al., 2025a).

- **TimeKAN:** Employs a kernel attention network to decompose the time series into frequency components and model each for improved long-term forecasting (Huang et al., 2025).

- **TimeMixer:** Utilizes a decomposable multiscale mixing module to integrate information across different temporal scales for more robust predictions (Wang et al., 2024b).

- **iTransformer:** Introduces an "inverted" transformer architecture that swaps the roles of queries, keys, and values to simplify and accelerate time series modeling (Liu et al., 2024c).

- **PatchTST:** Treats a time series as a sequence of fixed-length patches and applies transformer-based patch-wise modeling to capture long-range dependencies (Nie et al., 2023).

- **TimesNet:** Constructs a 2D temporal-variation representation and applies joint time–frequency convolutions to capture general patterns in time series data (Wu et al., 2023).

- **DLinear:** Decomposes the series into trend and seasonal components, fits each with a simple linear model, and then recombines them for forecasting (Zeng et al., 2023).

- **FEDformer:** Leverages frequency-enhanced decomposition within a transformer framework to efficiently model and forecast long-term periodic patterns (Zhou et al., 2022).

- **WPMixer:** This method is an MLP-based model that performs multi-resolution wavelet decomposition to generate time–frequency patches which are then embedded and mixed via lightweight MLP modules, efficiently capturing both local and global patterns for long-term time series forecasting (Murad et al., 2025).

- **TimeLLM:** This method is a reprogramming framework to repurpose LLMs for general time series forecasting with the backbone language models kept intact (Jin et al., 2024).

### 4.1.3. EVALUATION METRICS.

Mean Absolute Error (MAE) and Mean Squared Error (MSE) are adopted as the evaluation metrics (Zhou et al., 2022), which are defined as follows.

$$MAE = \frac{1}{M}\sum_{m=1}^{M}|\hat{y^m}-y^m|, \quad MSE = \frac{1}{M}\sum_{m=1}^{M}\|\hat{y^m}-y^m\|^2 \tag{19}$$

where $M$ is the testing data size, $\hat{y^m}$ is the prediction and $y^m$ is the ground truth. The smaller the MAE and the MSE are, the more accurate the method is.

### 4.1.4. IMPLEMENTATION DETAILS.

We implement our model using PyTorch on the NVIDIA A800 GPU. The hyperparameters in the model are set as

follows. The weight of $\mathcal{L}_{rep}$ at the representation level and $\mathcal{L}_{grad}$ at the gradient level are set to 1/8 and 1/2, respectively. The weight of loss at knowledge distillation is set to 0.001. The dropout rate in the decomposition block is set to 0.1. The patch length and stride are set to 16 and 8, respectively. The initial learning rate is 0.0001. ETT datasets and other datasets are split into the training data, validation data, and test data by the ratios of 6:2:2 and 7:1:2, respectively. The parameters of the baseline methods are set according to their original papers and any accompanying code. For the source-free training, we first train a source model on a source dataset (e.g., ETTh1). Then, we initialize the target model with the trained source model and then train the target model on the target domain (e.g., Weather). For example, we use ETTh1 $\rightarrow$ Weather to denote a source-free training process where the source and target domains are ETTh1 and Weather, respectively. To enable fair comparison, we train the baselines on the source data and then finetune them with 30% of the target data for testing. All baselines are evaluated under the strict source-free setting, where only the pretrained source model is available, and no source data is used during adaptation. The target domain data are used in the same manner as in our method, ensuring a fair comparison. We employ the stacked TSFE as the forecaster (Miao et al., 2024), and OFA as the LLM backbone. All baselines follow the same experimental setup with prediction length $PL \in \{96, 192, 336\}$ on all datasets.

### 4.2. Experimental Results

#### 4.2.1. OVERALL PERFORMANCE COMPARISON

We evaluate the source-free long-term forecasting capabilities of TimeID and baselines on six datasets (ETTh2, ETTm1, ETTm2, Weather, Electricity, Traffic) transferred from ETTh1 in Table 1. Results on other source datasets are provided in A.2.1. From the comparison results, it is evident that TimeID achieves the best performance on most datasets across all prediction lengths. Averaged across all 18 tasks (6 datasets $\times$ 3 prediction lengths), TimeID obtains the lowest average MSE and MAE, outperforming the most advanced method (Time-LLM and OFA) with an average MSE reduction by 4.98% and 4.39%, and MAE reduction by 2.64% and 3.21%, respectively. The most substantial improvements are observed on the ETTh1 $\rightarrow$ Weather and ETTh1 $\rightarrow$ ETTh2 datasets, particularly at shorter horizons (e.g., PL = 96), where TimeID reduces MSE by over 10.00% and MAE by over 5.03% compared to Time-LLM. This is due to the fact that TimeID learns the invariant features contained in time series through IDFL, and leverages LLM, denoised via proxy denoising, to guide the target model. Moreover, TimeID is particularly effective on complex datasets. On ETTh1 $\rightarrow$ Traffic dataset, where baseline methods suffer from high variance due to noise and periodicity. For example, at PL = 96, TimeID reduces MAE by 3.25% compared

to Time-LLM. The results demonstrate that TimeID has a superior generalization ability.

### 4.2.2. ABLATION STUDIES

To assess the contribution of each component in TimeID, we evaluate five variants: (1) *w/o_IDFL*: TimeID without the invariant disentangled feature learning module; (2) *w/o_LLM*: TimeID without the large language model; (3) *w/o_PD*: TimeID without the proxy denoising; (4) *w/o_KD*: TimeID without the knowledge distillation; and (5) *w/o_GI*: TimeID without the gradient invariant block, and report ablation study results in Figure 4 and Appendix A.2.2. The ablation results demonstrate that each component contributes meaningfully to the overall performance. The most substantial performance drop across most datasets is observed when knowledge distillation is removed. For example, on ETTh1 → ETTh2, the MSE rises by 17.86% and MAE rises by 18.34%. The reason is that knowledge distillation proves to be critical, as it effectively transfers knowledge from the LLM to the target model. The IDFL module proves to be equally vital, particularly in scenarios with complex distribution shifts. For instance, on ETTh1 → Traffic, removing IDFL causes the MSE and MAE to surge by 20.13% and 26.91%, respectively. In addition, excluding proxy denoising, the LLM or the gradient invariant block causes consistent but relatively moderate degradation, indicating that these components further stabilize and enhance adaptation performance.

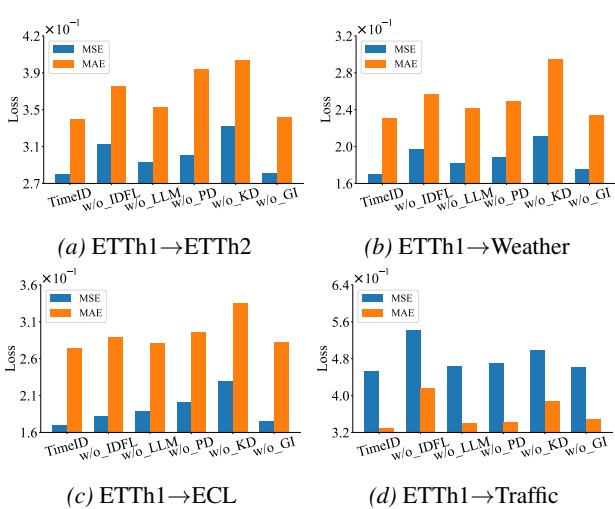

*(a)* ETTh1→ETTh2      *(b)* ETTh1→Weather

*(c)* ETTh1→ECL      *(d)* ETTh1→Traffic

*Figure 4.* Performance of TimeID and its variants.

### 4.2.3. EFFECT OF THE SIZE OF THE TARGET DOMAIN DATASET

To verify the scalability of TimeID, we conduct experiments using 5%, 10%, 20%, 30%, and 40% of the training set of the dataset. We observe that increasing the proportion of the target domain dataset generally improves performance in terms of MSE and MAE. Specifically, on the

ETTh1 → ETTh2 and ETTh1 → ETTm2 tasks, the model exhibits a consistent decrease in both MSE and MAE as more target domain data is used, as shown in Figure 5. This suggests that these tasks benefit significantly from domain-specific supervision and that the TimeID is capable of effectively leveraging more target samples. More experimental results about parameter sensitivity are reported in Appendix A.2.

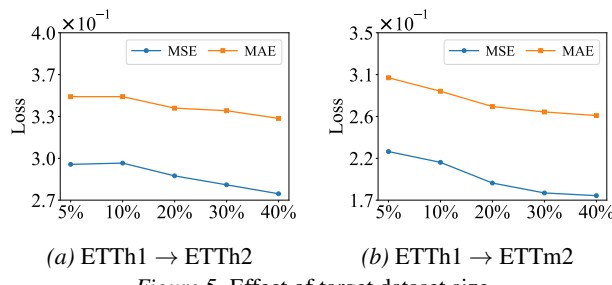

*(a)* ETTh1 → ETTh2      *(b)* ETTh1 → ETTm2

*Figure 5.* Effect of target dataset size.

### 4.2.4. DATA DISTRIBUTION VISUALIZATION OF PREDICTION AND GROUND TRUTH

To validate whether the prediction follows the similar distribution as the ground truth, we visualize the t-SNE (Maaten & Hinton, 2008) across ETTh1, Traffic, and ECL datasets, as shown in Figure 6. 1000 prediction–ground truth pairs are randomly sampled. Across all subfigures, the predicted values (orange) closely align with the true values (blue), indicating that TimeID effectively captures the underlying data distribution. Furthermore, we observe that the spatial structure of the data clusters is also well preserved between predictions and ground truth.

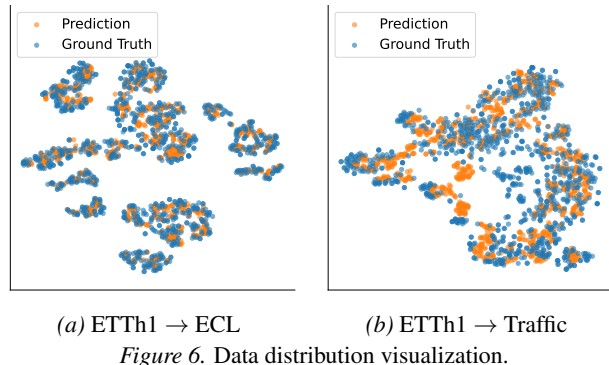

*(a)* ETTh1 → ECL      *(b)* ETTh1 → Traffic

*Figure 6.* Data distribution visualization.

### 4.2.5. CASE STUDY ON PREDICTION CONSISTENCY

To further intuitively demonstrate the effectiveness of the proposed TimeID, we compare the predictions with the ground truth on ETTh1 → ECL and ETTh1 → Traffic settings. As shown in Figure 7, the predicted curves closely follow the actual trajectories, capturing both periodic trends and abrupt variations. On the ECL, TimeID reproduces seasonal peaks and troughs with minor deviations at sharp transitions. On the Traffic dataset, the predictions remain well aligned with sudden spikes, demonstrating robustness

*Table 2.* Model results compared with time series foundation models in the 30% few-shot setting, with the lookback length set to 336 and forecast horizons set to 96. The best results for each task are highlighted in bold and the second-best results are underlined.

| Dataset | ETTh1 → ETTh2 | | ETTh1 → ETTm1 | | ETTh1 → ETTm2 | | ETTh1 → Weather | | ETTh1 → Electricity | | ETTh1 → Traffic | |
|---|---|---|---|---|---|---|---|---|---|---|---|---|
| Metric | MSE | MAE | MSE | MAE | MSE | MAE | MSE | MAE | MSE | MAE | MSE | MAE |
| Chronos | 0.295 | 0.339 | 0.405 | 0.397 | 0.192 | **0.256** | 0.190 | 0.229 | 0.216 | 0.294 | 0.399 | 0.251 |
| UniTS | 0.305 | 0.357 | 0.702 | 0.549 | 0.178 | 0.264 | 0.255 | 0.308 | 0.235 | 0.358 | 0.479 | 0.393 |
| MOIRAI | 0.300 | 0.341 | 0.528 | 0.435 | 0.229 | 0.299 | 0.223 | 0.237 | 0.175 | 0.290 | **0.368** | **0.219** |
| Moment | 0.290 | 0.347 | 0.466 | 0.462 | 0.188 | 0.275 | 0.406 | 0.346 | 0.394 | 0.447 | 0.407 | 0.290 |
| **TimeID (Ours)** | **0.280** | **0.338** | **0.359** | **0.390** | **0.177** | 0.267 | **0.169** | **0.228** | **0.170** | **0.276** | 0.452 | 0.327 |

across domains. These results qualitatively confirm that TimeID generalizes well and produces reliable forecasts beyond quantitative metrics.

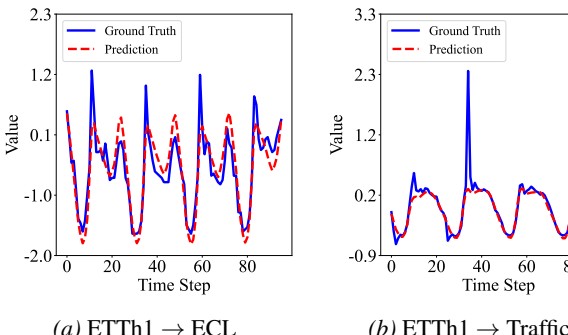

*(a)* ETTh1 → ECL        *(b)* ETTh1 → Traffic

*Figure 7.* Prediction vs. Ground Truth.

### 4.2.6. COMPARISON WITH FEW-SHOT OF TIME SERIES FOUNDATION MODELS

We further evaluate TimeID against time series foundation models, including Chronos (Ansari et al.), UniTS (Gao et al., 2024), MOIRAI (Woo et al., 2024), and Moment (Goswami et al., 2024), in a few-shot setting across six target datasets. As shown in Table 2, despite not having access to the massive multi-domain pretraining data used by these foundation models, TimeID achieves superior or highly competitive performance. For instance, TimeID yields the best MSE on ETTh2 and Weather, outperforming the best foundation model by 3.4% and 11.1%, respectively. This demonstrates that our proxy denoising and invariant feature learning can effectively bridge the domain gap, sometimes even outperforming general-purpose time series foundation models that lack domain-specific adaptation. We observe that on highly high-dimensional and complex datasets like Traffic, Chronos and MOIRAI maintain an advantage. This is likely due to the massive scale of their pre-training, which captures complex spatial-temporal dependencies that a single source-adapted model might struggle with. However, TimeID remains a more parameter-efficient and accurate alternative to large-scale time series foundation models in specialized source-free adaptation scenarios.

### 4.2.7. MODEL EFFICIENCY ANALYSIS

To evaluate the practical applicability of TimeID, we analyze its training efficiency in terms of training time per

*Table 3.* Training time, GPU memory cost and the number of parameters of TimeID.

| Dataset | Training Time | GPU Memory | Model Size |
|---|---|---|---|
| ETTh1 → ETTh2 | 19.51 s/epoch | 44.62 GB | 1.842 M |
| ETTh1 → ETTm1 | 81.62 s/epoch | 44.62 GB | 1.842 M |
| ETTh1 → ETTm2 | 79.90 s/epoch | 44.62 GB | 1.842 M |
| ETTh1 → ECL | 1970.30 s/epoch | 32.05 GB | 1.842 M |
| ETTh1 → Traffic | 3414.61 s/epoch | 42.97 GB | 1.842 M |
| ETTh1 → Weather | 261.48 s/epoch | 33.62 GB | 1.842 M |

epoch and peak GPU memory consumption, as reported in Table 3. Although TimeID adopts a dual-branch feature decoupling architecture and introduces an LLM as a proxy predictor, its adaptation process remains resource-friendly. The training time mainly depends on the sequence length and target data scale. Notably, the source model $\theta_s$ and the LLM proxy $\theta_{ts}$ are frozen during adaptation, ensuring that the computational overhead is not dominated by the LLM. In terms of memory usage, TimeID consistently consumes around 42–45 GB of GPU memory across most datasets, which is primarily caused by jointly loading the frozen source model, LLM proxy, and the trainable target model, along with intermediate feature representations. We also show the target model size in terms of the number of parameters. Overall, these results indicate that TimeID introduces a manageable training overhead and is practical for real-world deployment.

## 5. Conclusion

We present TimeID, a new source-free time series forecasting framework with proxy denoising that unleashes the power of LLMs and sufficient knowledge extracted from the source domain without accessing its raw data. To enable effective temporal correlation capturing and alleviate concept drift across domains, we propose an invariant disentangled feature learning module based on a dual-branch architecture. Further, a proxy denoising mechanism is proposed to dynamically incorporate the generalized knowledge learned by LLMs, enhancing model performance. We also employ the knowledge distillation to calibrate the final prediction with denoised prediction. An empirical study on real datasets offers evidence that the paper's proposals improve on the state-of-the-art in terms of prediction accuracy. An interesting research direction is to attempt to apply the proposed TimeID to other time series related tasks, e.g., classification.

## Acknowledgements

This work is partially supported by DFF Inge Lehmann grant (No. 4303-00014), National Natural Science Foundation of China (62372179) and SCRI, The Hong Kong Polytechnic University (No. Q-CDDG). Chenxi Liu is supported by the InnoHK funding.

## Impact Statement

This paper presents work whose goal is to advance the field of Machine Learning through a novel source-free time series forecasting framework. There are many potential societal consequences of our work, none of which we feel must be specifically highlighted here.

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

# A. Appendix

## A.1. Preliminary

This section introduces the basic concepts, notations, and preliminaries that underpin the proposed TimeID framework. We first formalize the source-free time-series forecasting problem under domain shift, and then establish a theory of invariant disentangled features for domain robustness, and introduce a proxy confidence theory for bias correction in large language models. Throughout, we adopt the following general notation: Bold lower-case symbols (e.g., $\mathbf{x}, \mathbf{y}$) denote vectors; bold upper-case symbols (e.g., $\mathbf{X}, \mathbf{Y}$) denote matrices or higher-order tensors. Calligraphic symbols (e.g., $\mathcal{X}, \mathcal{Y}$) denote sets or distributions. Subscripts $s$ and $t$ indicate source and target domains, respectively. Unless stated otherwise, all norms $\|\cdot\|$ are L2 norms.

### A.1.1. SOURCE-FREE TIME-SERIES FORECASTING

Let $\mathcal{T}$ denote a multivariate time-series sample of length $L$ and channel (feature) dimension $C$: $\mathcal{T} \in \mathbb{R}^{L \times C}$. Given a look-back window of length $l$: $\mathbf{x} = \mathcal{T}\{t - l + 1 : t, \cdot\} \in \mathbb{R}^{l \times C}$ the forecasting task is to predict the next $H$ steps: $\mathbf{y} = \mathcal{T}\{t + 1 : t + H, \cdot\} \in \mathbb{R}^{H \times C}$.

**Domains.** Source domain $\mathcal{D}_s = \{\mathcal{X}_s, \mathcal{Y}_s\}$ contains labeled pairs $(\mathbf{x}_s, \mathbf{y}_s)$ drawn from distribution $\mathcal{P}_s(\mathcal{X}, \mathcal{Y})$. Target domain $\mathcal{D}_t = \{\mathcal{X}_t\}$ contains unlabeled samples $\mathbf{x}_t$ from distribution $\mathcal{P}_t\{\mathcal{X}\}$, where $\mathcal{P}_t \neq \mathcal{P}_s$ .

**Source-free constraint.** At adaptation time, the original source data $\{\mathbf{x}_s, \mathbf{y}_s\}$ are inaccessible due to privacy or legal constraints. Only a pre-trained source model $\theta_s$ (parameterized by $\phi_s$) is available. The goal is to learn a target model $\theta_t$ (parameterized by $\phi_t$) that minimizes the expected forecast error on $\mathcal{D}_t$:

$$min_{\phi_t} \mathbb{E}_{\mathbf{x} \sim \mathcal{P}_t}[\mathcal{L}(\theta_t(\mathbf{x}), \mathbf{y})], \qquad (20)$$

where $\mathbf{y}$ is the (unknown) ground-truth future values, and $\mathcal{L}(\cdot, \cdot)$ is a loss function (e.g., MSE).

### A.1.2. INVARIANT DISENTANGLED FEATURES

Domain shifts in time series manifest as perturbations to trend $\mathcal{T}_{tre}$ or seasonality $\mathcal{S}_{sea}$. An invariant feature $\phi^*$ satisfies:

$$\phi^*(x) \approx \phi^*(x') \\ \forall x, x' \text{ s.t. } x \in D_i, x' \in D_j, C(x) = C(x') \qquad (21)$$

where $\mathcal{D}_i, \mathcal{D}_j$ are domains (e.g., differing trend contexts), and $\mathcal{C}$ denotes the component class (e.g., seasonal pattern). Disentanglement requires that features are component-specific: $\phi_s(x)$ encodes seasonality and is invariant to trend variations $\Delta t$, and $\phi_t(x)$ encodes trend and is invariant to

seasonal variations $\Delta s$. Formally, for small perturbations $\epsilon$:

$$\left\| \phi_s(s + t + \varepsilon \Delta t) - \phi_s(s + t) \right\|_2 < \delta_s, \\ \left\| \phi_t(s + t + \varepsilon \Delta s) - \phi_t(s + t) \right\|_2 < \delta_t, \qquad (22)$$

where $\delta_s, \delta_t \to 0$ for perfect invariance. Learning such features necessitates suppressing gradient pathways sensitive to cross-component variations, enabling generalization across domains where either component shifts (Parascandolo et al., 2021).

### A.1.3. PROXY CONFIDENCE THEORY

When a pre-trained large language model is used as a proxy forecaster on the unlabeled target domain, its outputs inevitably deviate from the latent "domain-invariant" distribution because of domain shift. We therefore treat the large language model as a noisy proxy and quantify its reliability through a proxy confidence theory.

**Notations.** $\mathcal{D}_S$ represents the source domain distribution (known only via the pretrained source model $\theta_s$). $\mathcal{D}_T^t$ represents target model $\theta_t$ distribution at adaptation step t. $\mathcal{D}_{TS}$ represents proxy (LLM $\theta_{ts}$) distribution. $\mathcal{D}_l$ represents latent domain-invariant distribution.

**Proxy Error.** Define the proxy error at step t as the expected divergence between the proxy and the ideal space:

$$e_t = \mathbb{E}_{x \sim \mathcal{D}_T}[D(\theta_{ts}(x), \theta_l(x))], \qquad (23)$$

where $D(\cdot, \cdot)$ is a distance in logit space. Since $\theta_l$ is inaccessible, we approximate $e_t$ by the disagreement between source and target models:

$$e_t \approx \mathbb{E}_{x \sim \mathcal{D}_T}[\|\theta_s(x) - \theta_t(x)\|_2]. \qquad (24)$$

Larger disagreement $\Rightarrow$ larger proxy error.

**Proxy Confidence Score.** We map the error to a confidence weight:

$$\mathcal{C}_t = exp(-e_t/\tau) \in (0, 1], \qquad (25)$$

with temperature $\tau > 0$ . At $t = 0$, $\theta_t \approx \theta_s \Rightarrow e_t \approx 0 \Rightarrow \mathcal{C}_t \approx 1$(high trust), As adaptation proceeds, $\theta_t$ drifts from $\theta_s \Rightarrow e_t$ grows $\mathcal{C}_t \downarrow$(reduced trust).

The proxy confidence theory thus provides an online, parameter-free mechanism to quantify and mitigate the noise inherent in large language model forecasts during source-free domain adaptation.

## A.2. Experiments

### A.2.1. OVERALL PERFORMANCE COMPARISON

In this section, we present the experimental results that compare TimeID with OFA using other datasets (except ETTh1) as source data. The experimental results show that our method can also achieve better prediction results when other datasets are used as source data.

*Table 4.* Overall Performance Comparison.

| Methods | Dataset | ETTm2→Traffic | | | Weather→Electricity | | | Electricity→ETTm2 | | | Traffic→Weather | | | ETTh2→Weather | | | ETTm1→ETTh1 | | |
|---|---|---|---|---|---|---|---|---|---|---|---|---|---|---|---|---|---|---|---|---|
| | PL | 96 | 192 | 336 | 96 | 192 | 336 | 96 | 192 | 336 | 96 | 192 | 336 | 96 | 192 | 336 | 96 | 192 | 336 |
| OFA | MSE | 0.455 | 0.469 | 0.503 | 0.171 | 0.197 | 0.271 | 0.185 | 0.242 | 0.293 | 0.239 | 0.254 | 0.299 | 0.221 | 0.288 | 0.299 | 0.482 | 0.502 | 0.498 |
| | MAE | 0.336 | 0.344 | 0.368 | 0.279 | 0.293 | 0.313 | 0.270 | 0.306 | 0.337 | 0.295 | 0.297 | 0.328 | 0.277 | 0.325 | 0.322 | 0.470 | 0.486 | 0.484 |
| TimeID | MSE | **0.453** | **0.469** | **0.496** | **0.170** | **0.189** | **0.211** | **0.180** | **0.229** | **0.281** | **0.229** | **0.236** | **0.271** | **0.168** | **0.287** | **0.271** | **0.456** | **0.474** | **0.441** |
| | MAE | **0.327** | **0.341** | **0.362** | **0.275** | **0.293** | **0.312** | **0.267** | **0.304** | **0.335** | **0.285** | **0.285** | **0.307** | **0.226** | **0.321** | **0.308** | **0.454** | **0.464** | **0.453** |

### A.2.2. ABLATION STUDY

Here, we present the experimental results on the remaining datasets, as shown in Figure 8.

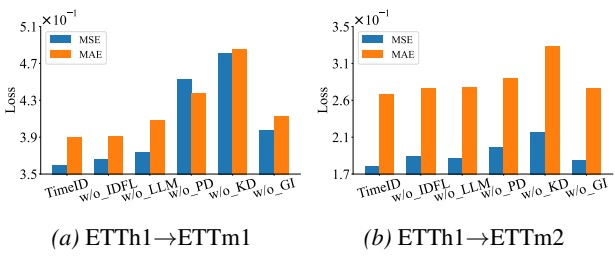

*(a)* ETTh1→ETTm1            *(b)* ETTh1→ETTm2

*Figure 8.* Performance of TimeID and its variants.

### A.2.3. INVARIANT FEATURE VISUALIZATION

To test whether TimeID is effective in capturing the invariant features, especially the seasonal and trend information, we visualize such frequency information of source data and target data in one figure with t-SNE. The results on two datasets are shown in Figure 9, where Figures 9 (a) and (b) show the season and trend comparison on the ETT dataset. Blue and orange dots represent the target data and source data, respectively. We observe that blue dots almost follow the trace of orange dots, indicating that the model learns the invariant features, i.e., season and trend information. Although some exceptions exist in the trend comparison, these show that TimeID not only learns the invariant features between source and target data but also is capable of extracting specialized features that are useful for the target time series forecasting.

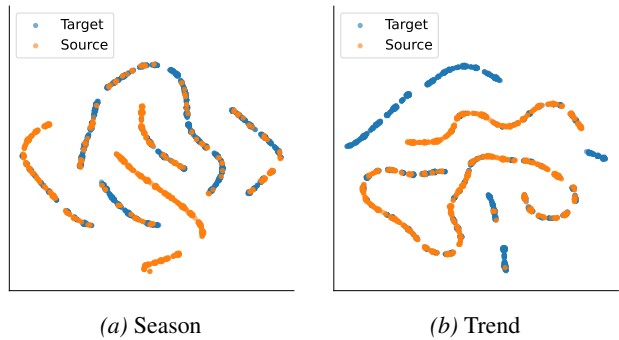

*(a)* Season            *(b)* Trend

*Figure 9.* Invariant feature visualization.

### A.2.4. EFFECT OF THE SIZE OF TARGET DOMAIN DATASET

Figure 10 shows experimental results on other datasets. In the main text, we mentioned that the degree of improvement varies significantly across datasets, and in some cases, additional data even leads to performance degradation. For example, the ETTh1 → ECL task shows very little variation across all data proportions, indicating that the model achieves near-optimal performance even with as little as 5% of target data. This insensitivity implies that the model transfers well to the ECL domain with minimal adaptation. Interestingly, ETTh1 → ETTm1 and ETTh1 → weather present non-monotonic trends. For example, in the ETTm1 task, performance initially worsens from 5% to 10% and then improves, while for weather, fluctuations occur throughout. This behavior may result from domain complexity, data noise, or overfitting due to insufficient generalization. For the ETTh1 → traffic task, model performance fluctuates within a narrow range across all data proportions. The lack of substantial improvement suggests that the model might have already captured the essential patterns with a small amount of target data, and further data adds limited value.

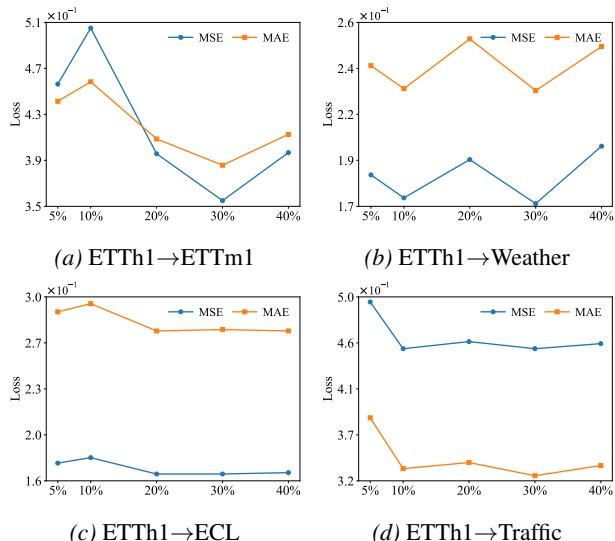

*(a)* ETTh1→ETTm1            *(b)* ETTh1→Weather

*(c)* ETTh1→ECL            *(d)* ETTh1→Traffic

*Figure 10.* Effect of target dataset size.

### A.2.5. EFFECT OF THE LEARNING RATE

We further study the sensitivity of our model to the learning rate, as shown in Figure 11. Across all transfer settings,

extremely small or large values lead to performance degradation, while moderate values (e.g., 1e-4) yield consistently better results. In particular, ETTh1→ETTm1 exhibits a sharp loss increase when deviating from this range, indicating that the model is relatively sensitive to the learning rate in certain domains. Overall, these results suggest that our method remains robust within a reasonable range but requires careful tuning to achieve optimal performance.

stability of our model.

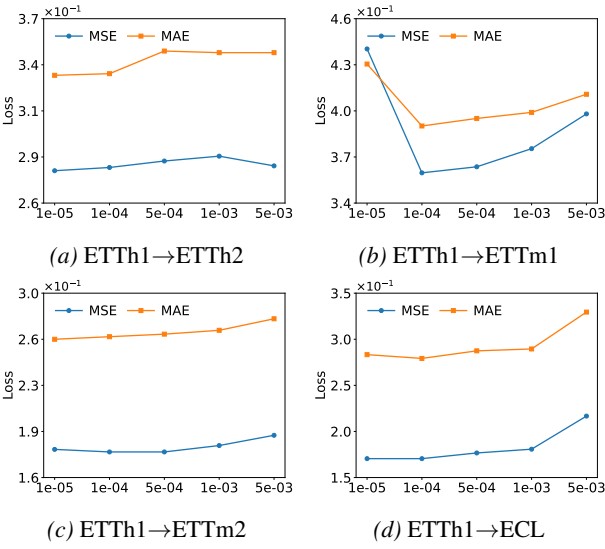

*Figure 11.* Effect of the learning rate.

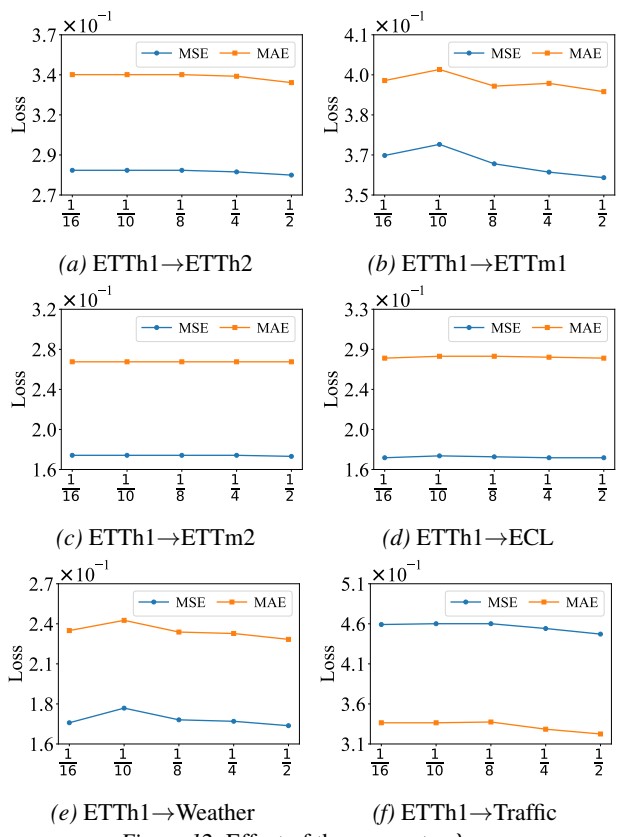

*(a)* ETTh1→ETTh2     *(b)* ETTh1→ETTm1

*(c)* ETTh1→ETTm2     *(d)* ETTh1→ECL

*(e)* ETTh1→Weather     *(f)* ETTh1→Traffic

*Figure 12.* Effect of the parameter $\lambda_{inv}$.

### A.2.6. EFFECT OF THE PARAMETER $\lambda_{inv}$

In this section, we analyze the sensitivity of the overall TimeID framework to the parameter $\lambda_{inv}$, which controls the contribution of the invariant feature learning objective in the total loss function (Eq. (18)). The weight $\lambda_{inv}$ is crucial as it balances the need to enforce domain invariance against the primary forecasting task. We evaluate TimeID using different settings for $\lambda_{inv} \in \{1/2, 1/4, 1/8, 1/10, 1/16\}$, and report the results in Figure 12. The empirical results demonstrate that TimeID is robust to the choice of $\lambda_{inv}$. Specifically, the optimal performance is consistently achieved at $\lambda_{inv} = 1/2$ across almost all datasets, as evidenced by the lowest MSE and MAE metrics. For instance, on the ETTh1 → ETTm1 task, $\lambda_{inv} = 1/2$ yields MSE = 0.359, while setting $\lambda_{inv} = 1/10$ results in a slight degradation to MSE = 0.371. Although the performance remains stable for smaller weights, a slight deterioration is observed. This slight degradation when $\lambda_{inv}$ is very small (e.g., $1/10$ or $1/16$) suggests that while the proxy denoising and knowledge distillation are powerful, a minimum degree of explicit invariant feature enforcement is necessary to effectively mitigate the domain shift. Conversely, setting $\lambda_{inv} = 1/2$ provides a sufficient regularization strength without overly constraining the target model's adaptation capabilities. Based on this analysis, we select the optimal parameter $\lambda_{inv} = 1/2$ for all main experiments, validating its effectiveness and the

### A.2.7. EFFECT OF THE PARAMETER $\lambda_{pred}$

Beyond the weight $\lambda_{inv}$, the overall performance of TimeID is also influenced by the loss weight $\lambda_{pred}$, which governs the contribution of the cross-domain prediction consistency loss $\mathcal{L}_{pred}$ (Eq. (12)) in the total objective (Eq. (18)). We examine the sensitivity of TimeID using $\lambda_{pred} \in \{1/2, 1/4, 1/8, 1/10, 1/16\}$ and present the results in Figure 14. The results show that TimeID exhibits considerable stability across most parameter settings, especially when $\lambda_{pred} \leq 1/4$. The empirical optimal performance is observed when $\lambda_{pred}$ is set to $1/4$ or $1/8$. For instance, on the ETTh1 → ETTm1 task, $\lambda_{pred} = 1/4$ yields the lowest MSE = 0.359 and MAE = 0.390, which is superior to the performance at $\lambda_{pred} = 1/2$ (MSE = 0.372). A clear performance degradation is noticeable when $\lambda_{pred}$ is too high (e.g., $\lambda_{pred} = 1/2$). This suggests that an overly aggressive enforcement of prediction consistency $\mathcal{L}_{pred}$ can potentially harm the adaptation process. High $\lambda_{pred}$ values might overly restrict the target model $\theta_t$ from learning the unique temporal patterns of the target domain, effectively freezing $\theta_t$ too close to the source model's behavior, leading to underfitting on target specific features. Based on this analysis, we choose $\lambda_{pred} = 1/4$ for the main experiments, striking a pragmatic balance between prediction alignment and adaptation flexibility.

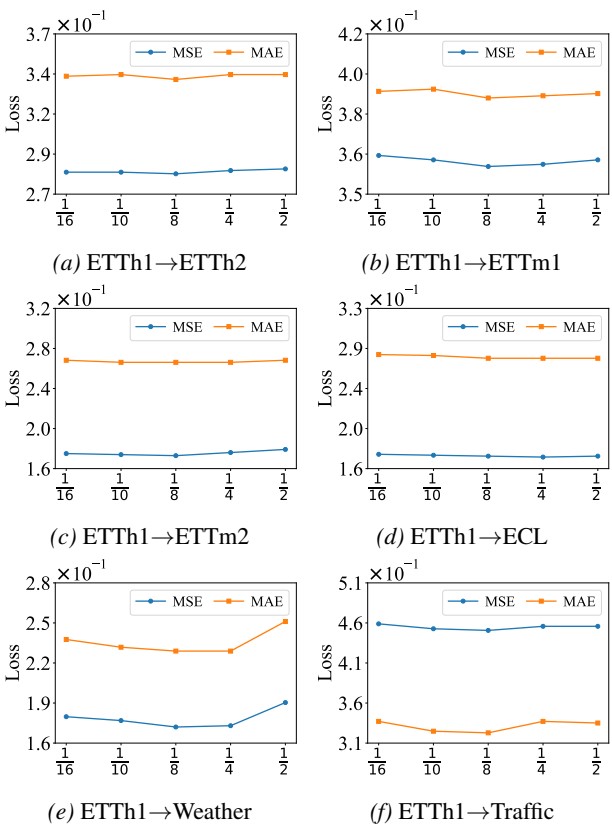

*Figure 13.* Effect of the parameter $\lambda_{rep}$.

firming the robust performance of the IDFL module across varying hyperparameter settings.

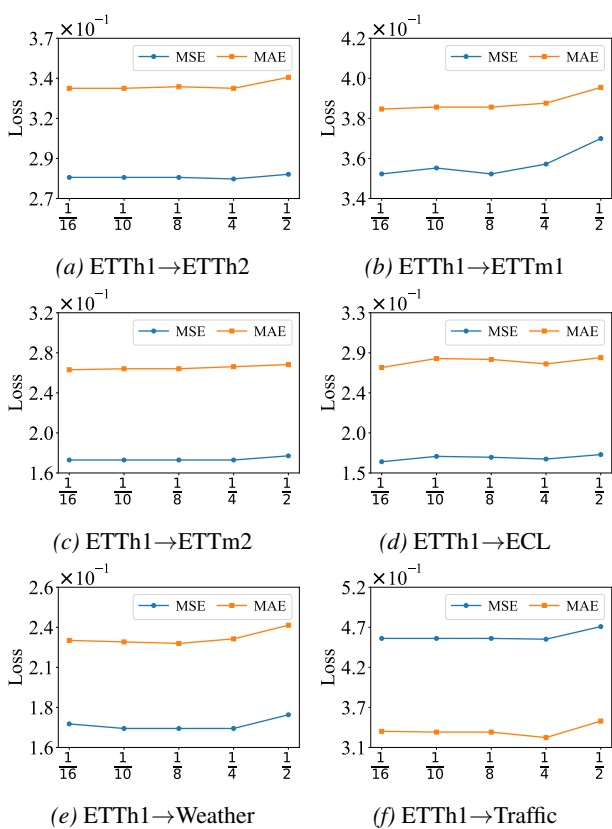

*Figure 14.* Effect of the parameter $\lambda_{pred}$.

### A.2.8. EFFECT OF THE PARAMETER $\lambda_{rep}$

We further investigate the robustness of TimeID to the representation-level invariant feature weight $\lambda_{rep}$, which is the balancing factor for the representation loss $\mathcal{L}_{rep}$ (Eq. (13)). We evaluate TimeID using various settings for $\lambda_{rep} \in \{1/2, 1/4, 1/8, 1/10, 1/16\}$, and report the results in Figure 13. The analysis reveals that TimeID is highly stable against variations in $\lambda_{rep}$ across the tested range, indicating that the core invariant features are effectively captured regardless of minor changes to the regularization strength. We observe the empirical best performance when $\lambda_{rep}$ is set to $1/8$. This setting achieves the lowest overall metrics, such as MSE = 0.452 and MAE = 0.327 on the challenging ETTh1 → Traffic task, and MSE = 0.169 on ETTh1 → Weather. This suggests that a moderate regularization strength is ideal for encouraging feature alignment without suppressing the model's ability to capture component-specific nuances. Performance remains strong and tightly clustered when $\lambda_{rep}$ is between $1/16$ and $1/4$. Although the overall degradation is minimal, setting $\lambda_{rep}$ too high (e.g., $1/2$) results in a slight drop in performance (e.g., MSE = 0.188 on Weather vs. 0.169 at $1/8$). An excessive weight on $\mathcal{L}_{rep}$ might over-constrain the feature space, potentially leading to a suboptimal balance between domain invariance and predictive accuracy. Based on these results, we adopt $\lambda_{rep} = 1/8$ for our main experiments, con-

### A.2.9. EFFECT OF THE PARAMETER $\lambda_{grad}$

We also conduct a sensitivity analysis on the gradient-level invariant feature weight $\lambda_{grad}$, which balances the contribution of the gradient loss $\mathcal{L}_{grad}$ (Eq. (15)) in the total objective. $\mathcal{L}_{grad}$ is designed to enforce a stringent constraint by aligning the gradients derived from the seasonal and trend features, ensuring a more robust feature decoupling that is resilient to domain shifts. We test $\lambda_{grad} \in \{1/2, 1/4, 1/8, 1/10, 1/16\}$ and report the outcomes in Figure 15. The results underscore the remarkable stability of TimeID across various $\lambda_{grad}$ values. The model consistently achieves strong performance, confirming that the dual branch IDFL design is inherently robust. The empirical best performance is consistently observed at $\lambda_{grad} = 1/2$. This setting yields the lowest metrics across most datasets, notably achieving MSE = 0.452 and MAE = 0.327 on the challenging ETTh1 → Traffic task. This indicates that a strong emphasis on gradient alignment is highly beneficial for learning truly component-invariant features, likely because gradient-level alignment imposes a stricter and more effective regularization than representation-level alignment. While the performance is stable, any deviation from $\lambda_{grad} = 1/2$ generally leads to a minor degradation. For

example, on the ETTh1 → Weather task, the MSE increases from 0.169 at $\lambda_{grad} = 1/2$ to 0.181 at $\lambda_{grad} = 1/4$. This suggests that while a wide range of weights is acceptable, maximizing the gradient-level constraint provides the best overall adaptation results. In summary, we select the optimal parameter $\lambda_{grad} = 1/2$ for all main experiments, validating the effectiveness of enforcing a strong gradient-level invariance constraint in source-free time series forecasting.

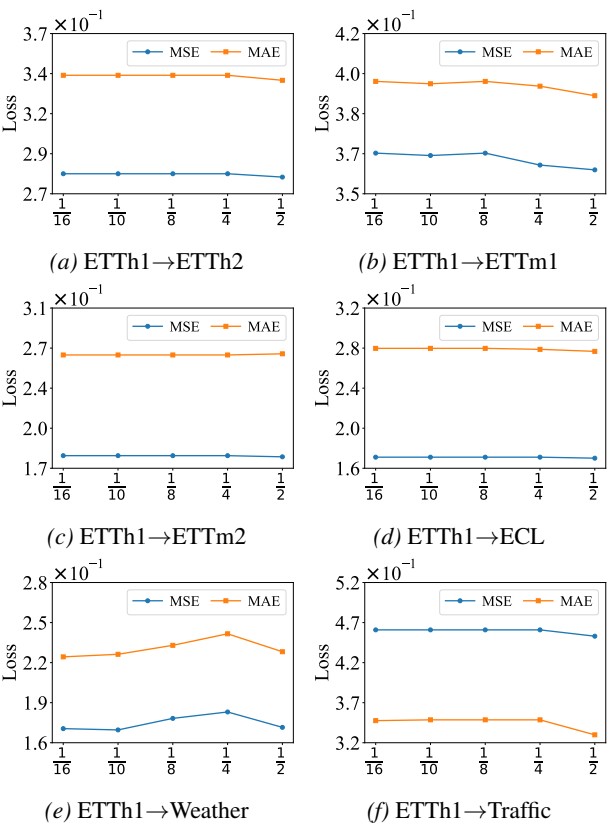

*Figure 15.* Effect of the parameter $\lambda_{grad}$.

### A.2.10. EFFECT OF THE PARAMETER $\lambda_{kd}$

We investigate the sensitivity of TimeID to the knowledge distillation weight $\lambda_{kd}$, which balances the influence of the denoised LLM proxy's supervision $\mathcal{L}_{kd}$ within the total objective (Eq. (18)). This parameter is vital as it dictates the extent to which the target model inherits cross-domain temporal knowledge from the large language model. We evaluate $\lambda_{kd}$ across a logarithmic scale $\in \{10^{-1}, 10^{-2}, 10^{-3}, 10^{-4}, 10^{-5}\}$ and summarize the results in Figure 16. The empirical results indicate that the performance of TimeID is highly sensitive but stable within an optimal range. The best forecasting accuracy is consistently achieved at $\lambda_{kd} = 10^{-3}$ across all datasets. For example, on the ETTh1 → ETTm1 task, $\lambda_{kd} = 10^{-3}$ results in an MSE = 0.359, whereas increasing the weight to $10^{-1}$ or decreasing it to $10^{-5}$ leads to a performance drop (MSE of 0.370 and 0.367, respectively). When $\lambda_{kd}$ is too

large, the target model may overly rely on the LLM's proxy predictions. Despite our denoising mechanism, an excessive distillation weight might introduce residual bias from the LLM, potentially suppressing the target model's ability to refine its own domain-specific representations. Conversely, when $\lambda_{kd}$ is too small, the "bridge" for knowledge transfer from the foundation model to the target model becomes too weak. In this case, the target model cannot fully leverage the rich temporal priors provided by the LLM, leading to suboptimal adaptation in data-scarce target domains. In conclusion, setting $\lambda_{kd} = 10^{-3}$ provides the most effective balance, ensuring that the target model sufficiently benefits from the LLM's generalization capabilities while maintaining its own adaptation flexibility.

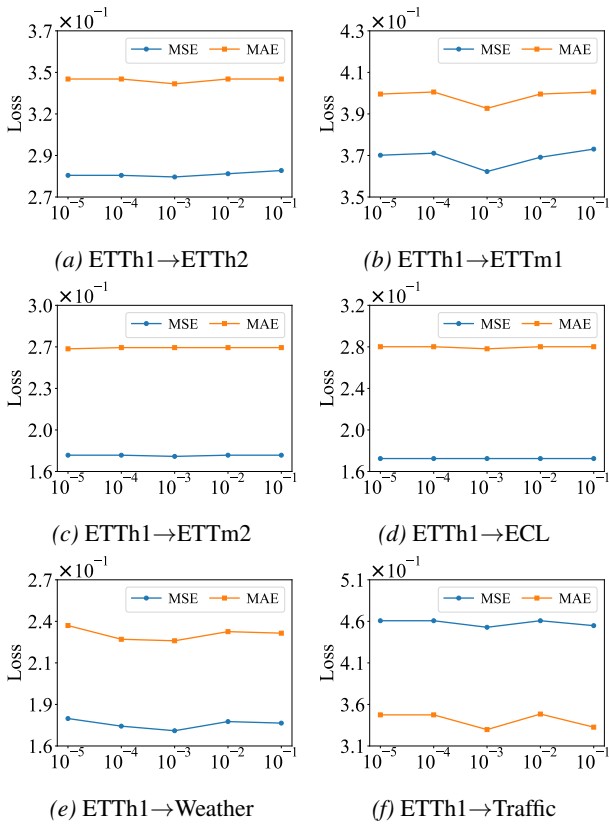

*Figure 16.* Effect of the parameter $\lambda_{kd}$.

### A.2.11. EFFECT OF THE CORRECTION STRENGTH $\alpha$

To further investigate the efficacy of the proxy denoising mechanism, we conduct a sensitivity analysis on the correction intensity parameter $\alpha$. As defined in Eq. (16), $\alpha$ modulates the degree to which the discrepancy between source and target model predictions is used to calibrate the LLM-based proxy forecasts. We evaluate $\alpha$ within the range $\in \{0, 0.25, 0.5, 0.75, 1\}$, where $\alpha = 0$ represents the baseline case with no denoising applied to the LLM outputs. The results are summarized in Figure 17. The empirical results demonstrate that TimeID consistently achieves its

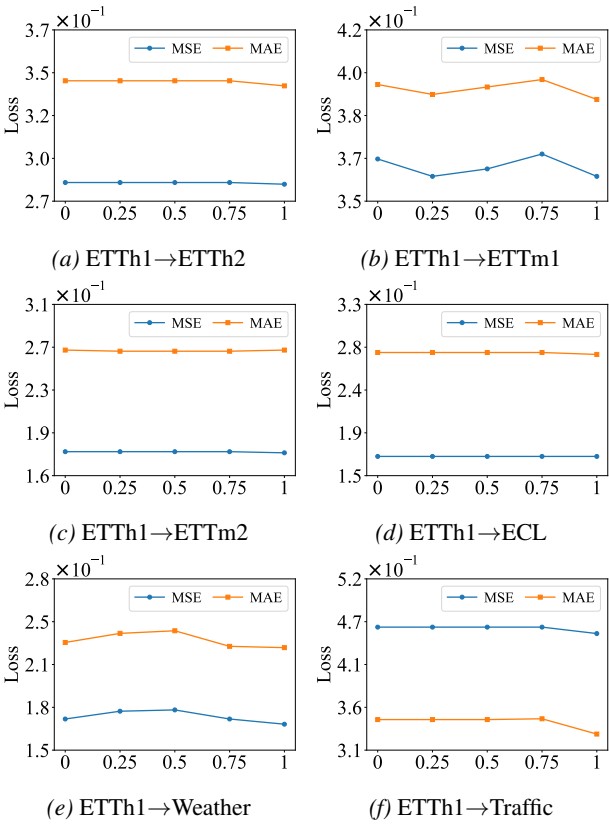

*Figure 17.* Effect of the correction strength $\alpha$.

peak performance at $\alpha = 1$ across all experimental scenarios. Specifically, on the ETTh1 $\rightarrow$ Traffic dataset, increasing $\alpha$ from 0 to 1 leads to a notable improvement, with MSE dropping from 0.460 to 0.452 and MAE from 0.345 to 0.327. A similar trend is observed in the ETTh1 $\rightarrow$ Weather task, where MSE reaches its minimum of 0.169 at $\alpha = 1$. Compared to the case without denoising ($\alpha = 0$), incorporating the correction term generally leads to superior or at least competitive results. This confirms our hypothesis that LLMs, while powerful, can exhibit systematic biases or hallucinations in specific time series domains that require dynamic calibration. The fact that the best performance is attained at $\alpha = 1$ suggests that the consensus between the frozen source model and the adapting target model provides a highly accurate error signal for denoising. A full strength correction effectively filters out domain-specific noise from the LLM proxy, providing the target model with a much cleaner and more reliable supervision signal during knowledge distillation. These observations validate the design of our proxy denoising module and justify the selection of $\alpha = 1$ as the default hyperparameter, ensuring the model's robustness and accuracy in source-free environments.

### A.2.12. EFFECT OF THE PERCENTILE $\alpha$

In the Invariant Disentangled Feature Learning (IDFL) module, we employ a percentile-based mask to identify and align

domain-invariant features while discarding domain-specific noise. We conduct a sensitivity analysis on this threshold parameter within the range $\in \{0.1, 0.3, 0.5, 0.7, 0.9\}$. This parameter controls the proportion of feature dimensions excluded during the representation-level alignment. The results across various datasets are illustrated in Figure 18. A critical observation is the sharp performance degradation on the ETTm1 dataset when the drop rate is set to 0.1, where the MSE surges to 1.676. This indicates that failing to sufficiently filter out domain-specific features allows non-transferable noise to contaminate the alignment process, leading to severe negative transfer in complex source-free scenarios. Beyond the extreme case of 0.1, the model exhibits high stability and strong performance across the range of 0.3 to 0.9. We observe that higher drop rates, such as 0.7 or 0.9, yield the best results for datasets like Traffic and ETTh2, suggesting that focusing on a core subset of highly invariant features is often beneficial. While higher values perform well in specific cases, we select 0.3 as the default setting for our main experiments. This choice strikes a balance between ensuring robust convergence on sensitive datasets like ETTm1 and maintaining high accuracy across other tasks, providing a safety margin for generalized source-free adaptation. In summary, this analysis highlights the necessity of feature filtering in representation-level alignment to prevent domain-specific noise from hindering the adaptation of invariant temporal patterns.

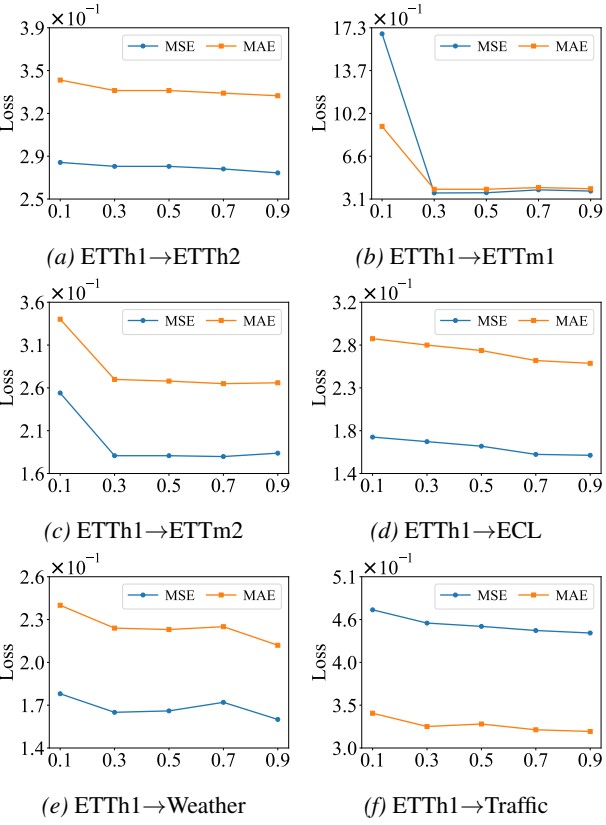

*Figure 18.* Effect of the percentile $\alpha$.

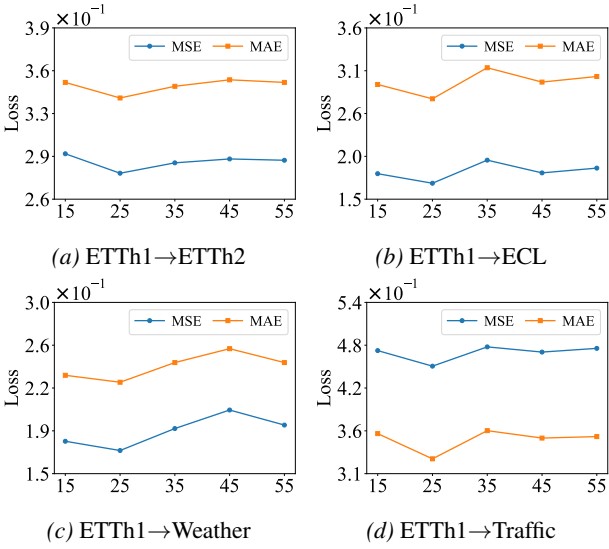

*Figure 19.* Effect of the moving average kernel length $k_{trend}$.

### A.2.13. EFFECT OF THE MOVING AVERAGE KERNEL LENGTH $k_{trend}$

The moving average kernel length $k_{trend}$ is a fundamental parameter in the series decomposition block of TimeID. It determines the window size used to extract the trend component from raw time series, which subsequently affects the disentanglement of seasonal features. We evaluate the impact of $k_{trend} \in \{15, 25, 35, 45, 55\}$ on model performance and report the results in Figure 19. The experimental results consistently indicate that $k_{trend} = 25$ is the optimal kernel length across all datasets. At this setting, the model achieves its lowest error rates. This suggests that a 25-step window provides an appropriate receptive field to capture the underlying trend without introducing excessive lag or over smoothing. We observe a significant performance degradation when the kernel length exceeds 25. Theoretically, an excessively large kernel leads to over smoothing, where high frequency but meaningful structural shifts in the trend are erroneously treated as seasonal noise, thereby hindering the IDFL module's ability to learn accurate invariant representations. Conversely, a smaller kernel also results in a slight performance loss. A window that is too narrow may fail to effectively filter out seasonal fluctuations, leaving residual oscillations in the trend component and complicating the decoupling process. Consequently, we fix $k_{trend} = 25$ as the default value to ensure a robust and precise decomposition of time series components, facilitating effective cross domain feature alignment.

### A.2.14. COMPARISON WITH OTHER DENOISING STRATEGIES

We conduct experiments against other denoising techniques by replacing the proxy denoising with moving average denoising, wavelet denoising and gaussian denoising. As shown in Table 5, the results show that proxy denoising outperforms other methods. The standard denoising method focuses on removing high-frequency random noise but cannot correct semantic forecasting shifts caused by domain gaps. The proposed proxy denoising is a knowledge-level correction rather than a signal-level filter. Therefore, it outperforms standard methods when the noise is systematic distribution shifts.

*Table 5.* Performance comparison with other denoising strategies.

| Method | Moving Average Denoising | | Wavelet Denoising | | Gaussian Denoising | | TimeID | |
|---|---|---|---|---|---|---|---|---|
| Metric | MSE | MAE | MSE | MAE | MSE | MAE | MSE | MAE |
| ETTh1 → Weather | 0.284 | 0.298 | 0.822 | 0.533 | 1.782 | 0.730 | **0.169** | **0.228** |
| ETTh1 → Electricity | 0.170 | 0.278 | 0.170 | 0.278 | 0.170 | 0.278 | **0.170** | **0.276** |
| ETTh1 → Traffic | 0.460 | 0.345 | 0.460 | 0.345 | 0.460 | 0.346 | **0.452** | **0.327** |

### A.2.15. CASE STUDY ON OPTIMIZATION PROCESS

We visualize the training loss curve on ETTh1 → ETTm1, as shown in Figure 20. The training loss curve demonstrates that the optimization process is robust and stable.

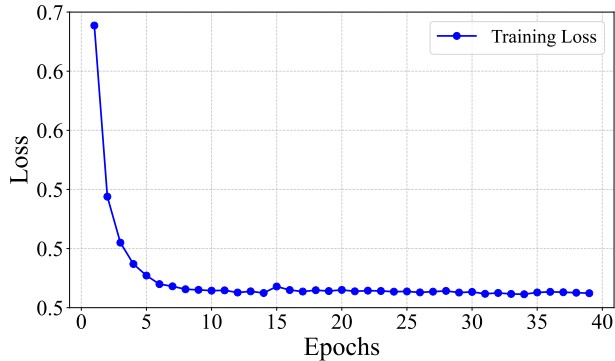

*Figure 20.* Training loss curve on ETTh1 → ETTm1.

*Table 6.* Performance comparison in the zero-shot adaptation setting.

| Method | iTransformer | | OFA | | PatchTST | | TimeID | |
|---|---|---|---|---|---|---|---|---|
| Metric | MSE | MAE | MSE | MAE | MSE | MAE | MSE | MAE |
| ETTh1 → Weather | **0.169** | **0.218** | 0.215 | 0.269 | 0.199 | 0.260 | 1.782 | 0.730 |
| ETTh1 → Electricity | **0.140** | **0.238** | 0.220 | 0.312 | 0.227 | 0.322 | 0.170 | 0.278 |
| ETTh1 → Traffic | **0.402** | **0.291** | 0.714 | 0.449 | 0.695 | 0.445 | 0.460 | 0.346 |

### A.2.16. PERFORMANCE COMPARISON IN THE ZERO-SHOT ADAPTATION SETTING

We conduct an experiment in the zero-shot adaptation setting by directly performing zero-shot inference on the target data using the source model trained with IDFL. As shown in Table 6, the results indicate that TimeID does not perform as well as in the original setting. This is because the method is specifically designed for few-shot scenarios. The core components (e.g., proxy denoising and knowledge distillation) operate during the adaptation training process. In the zero-shot setting, these modules are not applicable, leading to a performance decline of TimeID.

## A.2.17. PERFORMANCE COMPARISON ON MORE DATASETS

We conduct additional experiments on the Exchange, Solar, and PEMS datasets, as shown in Table 7. The results show that TimeID achieves consistent performance improvements across all datasets, further demonstrating the effectiveness of the proposed method.

*Table 7.* Performance comparison on the other three datasets.

| Method | iTransformer | | OFA | | PatchTST | | TimeID | |
|---|---|---|---|---|---|---|---|---|
| Metric | MSE | MAE | MSE | MAE | MSE | MAE | MSE | MAE |
| ETTh1 → Exchange | 0.122 | 0.249 | 0.123 | 0.263 | 0.111 | 0.232 | **0.102** | **0.227** |
| ETTh1 → Solar | 0.257 | 0.310 | 0.259 | 0.322 | 0.250 | 0.312 | **0.234** | **0.306** |
| ETTh1 → PEMS03 | 0.215 | 0.325 | 0.220 | 0.328 | 0.245 | 0.344 | **0.215** | **0.321** |

## A.2.18. DISCUSSION ON ADAPTING THE PROPOSED METHOD TO TS FOUNDATION MODEL SCENARIOS

The framework is designed to be model-agnostic and can be naturally extended to scenarios involving time series foundation models. In particular, the LLM-based proxy model does not rely on a specific type of predictor, and can be replaced by TS foundation models. We conduct additional experiments using the time series foundation model (e.g., Chronos (Ansari et al.)) to replace the LLM-based time series forecasting model. As shown in Table 8, we find that performance declines when the LLM is replaced with the time series foundation model in most cases. This may be because TS foundation models (e.g., Chronos) are designed to capture general temporal patterns, but their capability to provide robust cross-domain transferable knowledge remains limited, which shows that LLMs are more suitable for source-free time series forecasting.

*Table 8.* Performance comparison between TSFM-based proxy model and LLM-based proxy model.

| Method | TSFM-based proxy model | | LLM-based proxy model (TimeID) | |
|---|---|---|---|---|
| Metric | MSE | MAE | MSE | MAE |
| ETTh1 → ETTh2 | 0.301 | 0.354 | **0.280** | **0.338** |
| ETTh1 → ETTm1 | 0.439 | 0.432 | **0.359** | **0.390** |
| ETTh1 → ETTm2 | 0.181 | 0.272 | **0.177** | **0.267** |
| ETTh1 → Weather | 1.465 | 0.700 | **0.169** | **0.228** |
| ETTh1 → Electricity | **0.169** | 0.276 | 0.170 | **0.276** |
| ETTh1 → Traffic | 0.461 | 0.338 | **0.452** | **0.327** |

*Table 9.* Performance comparison with domain adaptation methods.

| Method | ADvSKM | | CoDATS | | SASA | | TimeID | |
|---|---|---|---|---|---|---|---|---|
| Metric | MSE | MAE | MSE | MAE | MSE | MAE | MSE | MAE |
| ETTh1->weather | 0.241 | 0.332 | 0.570 | 0.579 | 0.565 | 0.576 | **0.169** | **0.228** |

## A.2.19. PERFORMANCE COMPARISON WITH DOMAIN ADAPTATION METHODS

We conduct additional experiments against more domain adaptation methods (i.e., ADvSKM (Liu & Xue, 2021),

CoDATS (Wilson et al., 2020), and SASA (Cai et al., 2021)), as shown in Table 9. The results show that TimeID achieves SOTA performance.

## A.2.20. PSEUDO TRAINING CODE OF TIMEID AND THE IDFL MODULE

We provide the pseudo training code of TimeID and the IDFL module in Algorithm 1 and Algorithm 2, respectively. In Algorithm 1, the source model is first trained on the source dataset and then used to initialize the target model. During source-free adaptation, lines 1-5 sample a mini-batch target data and obtain predictions from the frozen source model, the trainable target model, and the frozen LLM proxy. Lines 6-7 extract invariant seasonal and trend features through IDFL and compute the corresponding losses. Lines 8-9 perform proxy denoising by correcting the LLM proxy forecast according to the discrepancy between the source and target predictions. Lines 10-11 apply knowledge distillation between the denoised proxy prediction and the target prediction. Finally, line 12 updates the target model by minimizing the overall objective in Eq. (18).

---

**Algorithm 1** The TimeID Framework

---

**Input:** Source model $\theta_s$, Pre-trained large language model $\theta_{ts}$, Source dataset $X_S$, Target dataset $X_T$, denoising strength $\alpha$, loss weights $\lambda_1, \lambda_2$, iterations $M$.
**Output:** Adapted target model $\theta_t$.
**Initialization:** Train $\theta_s$ on source dataset $X_S$ and set target model $\theta_t \leftarrow \theta_s$.

1: **for** $m = 1$ to $M$ **do**
2:     Sample a mini-batch $X_T^b$ from $X_T$.
3:     Obtain source prediction $z_s$ by forwarding $X_T^b$ through $\theta_s$ (frozen).
4:     Obtain target prediction $z_t$ by forwarding $X_T^b$ through $\theta_t$.
5:     Obtain proxy forecast $z_{proxy}$ by forwarding $X_T^b$ through $\theta_{ts}$ (frozen).
6:     Extract invariant trend and seasonal features $h_{trend}, h_{season}$ via invariant feature learning.
7:     Compute invariance regularizers $L, L_{inv}, L_{pred}, L_{rep}, L_{grad}$ at representation and gradient levels (Eq. (3), Eq. (8), Eq. (12), Eq. (13), Eq. (15)).
8:     Apply proxy denoising to correct proxy forecasts of $X_T^b$ (Eq. (16)).
9:     $z_{denoised} \leftarrow z_{proxy} - \alpha(z_s - z_t)$.
10:     Apply knowledge distillation between corrected proxy $z_{denoised}$ and target outputs $z_t$ (Eq. (17)).
11:     $L_{kd} \leftarrow MSE(z_{denoised}, z_t)$.
12:     Compute the overall objective $L_{all}$ and update $\theta_t$ by minimizing $L_{all}$ (Eq. (18)).
13: **end for**
14: **return** adapted target model $\theta_t$

---

Algorithm 2 describes the training process of IDFL. Lines 1-4 decompose the input time series into seasonal and trend components and feed them into the corresponding forecasting branches. Lines 5-9 compute gradient attributions, construct invariant masks, and extract invariant seasonal and trend representations. Lines 10-12 generate the final prediction and compute the forecasting, representation-level invariance, and gradient-level invariance losses. Finally, lines 13-15 update the model parameters and return the trained IDFL module.

---

**Algorithm 2** Invariant Disentangled Feature Learning (IDFL)

---

**Input:** Training dataset $D = \{(x_i, y_i)\}_{i=1}^N$, decomposition function $Dec(\cdot)$, residual predictor $f_r$, trend predictor $f_t$, loss weights $\lambda_{rep}, \lambda_{grad}, \lambda_{inv}$.
**Output:** Trained model parameters $\theta$.
**Initialization:** Initialize model parameters $\theta$.

1: **for** each training iteration **do**
2:    Sample a mini-batch $(x, y)$ from dataset $D$.
3:    Decompose the input time series into residual and trend components $(x_{res}, x_{trend})$ using $Dec(\cdot)$ (Eq. (1)).
4:    Obtain residual and trend predictions by forwarding the decomposed inputs through predictors $f_r$ and $f_t$ (Eq. (2)).
5:    Perform a second forward pass to obtain $(\hat{y}_{res}^2, \hat{y}_{trend}^2)$ for gradient attribution.
6:    Compute gradient attributions of predictions with respect to the input (Eq. (4)).
7:    Estimate gradient discrepancies between the two forward passes (Eq. (5)).
8:    Construct invariant masks $M_{res}$ and $M_{trend}$ using a thresholding operation on the gradient differences (Eq. (6)).
9:    Extract invariant residual and trend representations $z_{res}$ and $z_{trend}$ by applying the masks to the input.
10:    Predict the final output $\hat{y} = f_r(z_{res}) + f_t(z_{trend})$.
11:    Compute training objectives including prediction loss $L_{pred}$, invariant features prediction loss $L_{inv}$, representation invariance loss $L_{rep}$, and gradient invariance regularization $L_{grad}$ (Eq. (3), Eq. (8), Eq. (12), Eq. (13), Eq. (15)).
12:    Compute the total loss $L = L_{pred} + \lambda_{inv}L_{inv} + \lambda_{rep}L_{rep} + \lambda_{grad}L_{grad}$.
13:    Update model parameters $\theta$ by minimizing the total loss.
14: **end for**
15: **return** trained model parameters $\theta$

---

