# OpenReview forum: "Invariant Representation Learning for Source-Free Time Series Forecasting with LLM-Centric Proxy Denoising"
_ICML.cc/2026/Conference — ICML 2026 regular_

### Official Review · Reviewer_pexZ · 2026-03-05

**Soundness:** 3
**Presentation:** 4
**Significance:** 3
**Originality:** 4
**Overall Recommendation:** 5
**Confidence:** 5

**Summary:**

This paper introduces TimeID, a source-free domain adaptation framework for time series forecasting. Unlike traditional transfer learning methods that accessing to both source and target data, TimeID transfers a pretrained source model to the target domain without requiring the source data. The framework combines three main ideas: (1) Invariant Disentangled Feature Learning (IDFL): a dual-branch seasonal/trend decomposition with both representation- and gradient-level explicit invariance, enhanced with frequency-consistent supervision via Fourier module. (2) Proxy Denoising (PD): treats an LLM as a noisy forecaster and dynamically corrects its systematic bias using source–target model consensus. (3) Knowledge Distillation (KD): transfers denoised, LLM-guided knowledge into a lightweight target model.

**Compliance With Llm Reviewing Policy:**

Affirmed.

**Final Justification:**

Thank you for the further clarification, which has addressed my concerns combining the rebuttal. As a result, my understanding of the paper has improved, and I feel more positive about the work. Thus, I recommend acceptance.

**Key Questions For Authors:**

A practical question, how robust is TimeID to addressing distribution shifts? Can TimeID work with extremely limited target data in real world scenarios?

**Limitations:**

yes

**Strengths And Weaknesses:**

### Strengths:

- This paper is overall well-written and easy to follow. The organization of this paper is clear and in good logic, making the paper understandable to readers.
- The paper addresses an important real world constraint: transferring source knowledge to target time seire for forecasting, which does not require access to original source data. By focusing on this source-free setting, the paper tackles a realistic and challenging problem that is not well addressed by existing time series forecasting methods.
- The framework is creative and well motivated combining disentangled trend/seasonal invariance, LLM-guided proxy denoising, and bidirectional knowledge transfer. Each component plays a clear role in the pipeline. The integration of these components is thoughtful and aligned with the goal of improving adaptation performance under source-free constraints.
- The decomposition-based structure provides interpretable insights (trend and seasonal components). This design helps the model focus on domain-invariant temporal patterns rather than domain-specific noise or fluctuations. It makes the method easier to understand and easier to diagnose when performance varies across datasets.



### Weaknesses:

- Distribution shift seems to be one of the main concerns of TimeID. More discussions are required to be provided about how the distribution shift can be addressed by the proposed framework and whether there are specifc designs.
- The model training via invariant disentangled feature learning is well described in the manuscript. A short high-level summary or pseudo code block that outlines how TimeID can achieve invariant feature learning would make this paper easier to follow.
- The experiments show that PatchTST performs better than OFA in terms of ETTh1 -> ETTh2 in most cases. It is requried to provide more explanations about this observation.

---

> ### Author Rebuttal · Authors · 2026-03-31
>
> Thank you for the constructive suggestions.
>
> **W1: The handling of the distribution shift in the proposed framework is insufficiently discussed.**
>
> Our framework addresses distribution shifts through three specific design components. First, the Invariant Disentangled Feature Learning (IDFL) module decomposes time series into trend and seasonal components and learns invariant representations that remain consistent across these components, which encourages the model to capture domain-invariant temporal patterns rather than domain-specific variations. Second, the proxy denoising mechanism reduces the effect of noisy proxy predictions caused by domain mismatch, improving the reliability of the supervisory signal during adaptation. Third, we establish a bidirectional knowledge transfer loop through a knowledge distillation module, where denoised proxy forecasts supervise the target model, while target predictions feed back to stabilize the proxy correction, preventing distribution drift from the target domain. We will further discuss the handling of distribution shift in the Methodology section.
>
> **W2: A high-level summary or pseudocode for invariant feature learning is needed.**
>
> We will include a concise pseudo-code block, as shown in Algorithm https://bashify.io/i/3KOv6G, summarizing the main training procedure of IDFL in the revised version.
>
> **W3: More explanation is needed for why PatchTST outperforms OFA in ETTh1 -> ETTh2.**
>
> PatchTST is specifically designed for time series forecasting and benefits from patch-based representations that capture local temporal dependencies effectively. In contrast, OFA is a general LLM-based model trained across multiple domains and tasks. OFA shows strong generalization ability, but its architecture is not optimized specifically for time series forecasting tasks. As a result, PatchTST may achieve better performance in certain forecasting scenarios such as ETTh1 → ETTh2. We will explain the reason in the Experiments section.
>
> **Q1: How robust is TimeID to addressing distribution shifts? Can TimeID work with extremely limited target data?**
>
> TimeID is designed to improve robustness under distribution shifts through invariant feature learning and proxy-guided supervision. Our experiments in Appendix A3.1 show that TimeID performs well in various adaptation tasks between different domains, which demonstrates that TimeID is robust to addressing distribution shifts. We conduct a sensitivity analysis experiment of the target dataset size using 5%, 10%, 20%, 30%, and 40% of the dataset in Section 4.2.3 of the paper. We can see that performance degrades when only using 5% of the dataset, but it still maintains within a stable range, which shows that TimeID can work with extremely limited target data. We will highlight the robustness of TimeID in the Methodology section.

---

> > ### Author Rebuttal · Reviewer_pexZ · 2026-04-01
> >
> > Thank you for the response, which addressed my previous questions adequately. I still have a very minor question. The authors do not clearly illustrate that if they run the baselines in source-free forecasting settings. More explanation regarding baseline settings are suggested to improve fairness and transparency.

---

> > > ### Author Response · Authors · 2026-04-01
> > >
> > > Thank you for the positive feedback.
> > >
> > > We would like to clarify the baseline settings as follows. We transform the original baselines to their source-free settings, i.e., all baselines are adapted to follow the source-free forecasting setting in experiments. In the source-free forecasting setting, the model has no access to source-domain data during adaptation, and only a pretrained model (or its parameters) and target-domain data are available. This setting reflects practical scenarios where source data cannot be shared due to privacy or storage constraints. In this paper, we first train them on the source data to get source models. Then, during adaptation, we only use the source models to train on a part of the target data (i.e., 30%), which do not have access to the source data during adaptation. We will add the baseline settings in the final version.

---

### Official Review · Reviewer_mHjN · 2026-03-11

**Soundness:** 2
**Presentation:** 2
**Significance:** 2
**Originality:** 2
**Overall Recommendation:** 3
**Confidence:** 5

**Summary:**

This paper studies source-free time series forecasting, where a model trained on a source domain must be adapted to a target domain without access to the original source data. To address this setting, the authors propose a framework that combines representation learning and LLM-based supervision. Specifically, the method first learns disentangled representations that separate domain-invariant and domain-specific temporal patterns in order to mitigate distribution shift between domains. To compensate for limited target-domain data, the framework leverages LLM-generated proxy signals as additional supervision. Since these signals may be noisy, the authors introduce a proxy denoising mechanism to filter unreliable predictions. The knowledge from the proxy model is then transferred to the forecasting model through knowledge distillation and gradient-level alignment. Experiments on several benchmark time series datasets are conducted to evaluate the proposed method and compare it with existing forecasting models.

**Compliance With Llm Reviewing Policy:**

Affirmed.

**Final Justification:**

My final recommendation remains unchanged after considering both the paper and the authors’ rebuttal. I appreciate that the paper studies a relevant problem—source-free time series forecasting—and proposes an interesting combination of invariant representation learning and LLM-based guidance. I also acknowledge the empirical effort in the paper, and the rebuttal improves the experimental positioning by adding comparisons to domain adaptation methods.

In terms of **originality** and **significance**, I see the work as promising: the problem setting is worthwhile, and the empirical gains suggest potential practical value. However, in terms of **soundness**, I still have important reservations. The rebuttal improves the framing of the challenges, but it does not fully resolve my main concerns about why LLMs are necessary in this setting, whether the invariance assumption is sufficiently justified under cross-domain temporal shift, and why the proposed proxy denoising mechanism should reliably estimate and correct LLM error.

In terms of **clarity**, the rebuttal is helpful and constructive, but the overall method still feels somewhat over-composed, and the core design principle is not yet fully clear to me. Overall, while the rebuttal addressed some presentation- and comparison-related concerns, it did not materially change my evaluation of the main conceptual issues. Therefore, I keep my original recommendation. I hope these comments help the authors strengthen the motivation, assumptions, and methodological justification in a future revision.

**Key Questions For Authors:**

See weaknesses above

**Limitations:**

Yes

**Strengths And Weaknesses:**

Strengths:

1. The paper focuses on source-free time series forecasting, where the source data are unavailable during adaptation. This setting is relevant in scenarios where data sharing is restricted due to privacy or storage constraints, and exploring solutions under this constraint is potentially useful.

2. The proposed framework integrates multiple techniques, including representation disentanglement, proxy-based supervision, and knowledge distillation. This combination provides an interesting attempt to leverage additional knowledge sources (e.g., LLM-generated signals) to assist adaptation under limited target data.

3.The empirical evaluation covers multiple commonly used time series forecasting datasets, which helps demonstrate the behavior of the proposed approach across different scenarios.


Weaknesses

1. The introduction lacks a clear logical transition explaining why large language models are necessary for the source-free time series forecasting problem. The paper first motivates the source-free setting, then introduces LLM-based forecasting methods, and later returns to the original problem formulation. This creates a somewhat discontinuous narrative, making it unclear how LLMs naturally arise as the appropriate solution for the proposed task.

2. The three challenges described in the introduction—target data scarcity, distribution shift, and LLM hallucination—are fairly general issues commonly encountered in machine learning. The paper does not sufficiently articulate what makes source-free time series forecasting uniquely difficult, nor does it provide deeper analysis of how these challenges manifest specifically in temporal forecasting tasks (e.g., shifts in temporal dynamics, seasonal structures, or trend patterns).

3. The method relies on invariant/domain-specific feature disentanglement to address source–target distribution shift. However, this approach implicitly assumes that the source and target domains share stable invariant temporal dynamics. The paper does not provide sufficient justification for this assumption, nor does it analyze the nature of distribution shift in time series data. If temporal dynamics themselves change across domains, invariant representations may not exist, weakening the rationale behind this design.

4. The framework introduces proxy signals generated from LLM forecasts and applies proxy denoising before distilling the information to the forecasting model. However, the paper does not clearly explain why such an intermediate proxy representation is necessary. Conceptually, the proxy signals appear similar to pseudo-label supervision, yet the distinction from existing pseudo-labeling approaches is not clearly articulated. Moreover, it remains unclear why the proposed denoising procedure can reliably correct errors in LLM-generated forecasts.

5. The paper focuses on a transfer-learning-style problem (source-free forecasting), but the experiments primarily compare against standard forecasting models such as TimesNet, PatchTST, and DLinear. These models are not designed for cross-domain adaptation. Without comparisons to transfer-learning or domain-adaptation methods for time series forecasting, it is difficult to assess whether the proposed approach truly advances the state of the art for the intended problem setting.

6. The proposed framework combines multiple modules, including Fourier-based feature decomposition, invariant/domain-specific disentanglement, LLM-based proxy generation, proxy denoising, knowledge distillation, and gradient-level alignment. While each component may have individual motivation, the paper does not clearly demonstrate why all of them are necessary or how they jointly contribute to solving the problem. As a result, the method appears somewhat over-engineered, and the absence of clear design principles makes it difficult to identify the core innovation.

7. The paper would benefit from improvements in writing and presentation clarity. Some parts of the introduction and method description are difficult to follow, and the logical connections between different components of the framework are not always clearly explained. For example, the motivation for certain modules (e.g., proxy denoising and gradient-level alignment) could be articulated more explicitly. In addition, several technical details appear scattered across sections, making it harder for readers to understand the overall design and how different components interact. Improving the clarity of the presentation and providing a more structured explanation of the framework would help readers better understand the proposed approach.

---

> ### Author Rebuttal · Authors · 2026-03-31
>
> Thanks for the insightful comments.
>
> **W1: Necessity of LLMs in source-free time series forecasting (SF-TSF).**
>
> Unlike traditional domain adaptation, which requires few source training samples, we study a novel problem of SF-TSF without accessing source data. To alleviate target data sparsity, we utilize LLMs to facilitate SF-TSF inspired by their powerful generalization capabilities. LLMs are considered an external knowledge source, enhancing model performance.
>
> **W2: Challenge clarification.**
>
> We would like to clarify that the three challenges are interrelated to the studied SF-TSF and will refine these challenges as follows. First, the time series can be scarce due to the data collection mechanism and sensor failure, resulting in **insufficient historical observations (Refined Challenge I)**. Second, the source and target time series may exhibit distribution differences, showing significant **temporary dynamic shift (Refined Challenge II)**, e.g., different trends and seasonalities. Third, LLMs are employed to guide the model training, they may struggle with hallucination, leading to **biased prediction guidance (Refined Challenge III)**.
>
> **W3: Rationality of the invariance assumption.**
>
> In this paper, we follow existing studies [1], which show domain-invariant features exist across target and source time series, enabling powerful representation learning. Further, we visualize the learned invariant features on ETT datasets for examples (see Figure https://bashify.io/i/Vf4gin), demonstrating the existence of invariant features. In addition, we provide a case study to show the distribution shift between different time series datasets (i.e., ETTh1 and Weather) with t-SNE. Figure https://bashify.io/i/G37dUu shows that a distribution difference exists between different time series datasets, e.g., ETTh1 and Weather. By disentangling trend and seasonal components, we extract invariant features in high-dimensional spaces instead of the original data, which will not be changed even if the temporal dynamic changes [2].
>
> [1]. Time Series Domain Adaptation Via Latent Invariant Causal Mechanism, TPAMI 2025.
>
> [2]. MDLR: A multi-task disentangled learning representations for unsupervised time series domain adaptation, IPM 2024.
>
> **W4: Necessity of Proxy Representation and Denoising.**
>
> We learn the proxy representation via an LLM-based TSF model, serving as an external knowledge base to guide the target model training, alleviating data scarcity.
>
> The learned proxy differs from pseudo-labeling as follows: (1) It is an intermediate feature integrating LLM and source–target information, rather than direct supervision. (2) It is dynamically updated via Proxy Denoising (PD) instead of being fixed. (3) It connects source and target models and LLMs, while the pseudo-labeling only relates to the target model.
>
> PD aims to reduce the prediction discrepancies between source and target models, enabling consistent temporal feature extraction. Combining with knowledge distillation, TimeID can correct errors reliably.
>
> **W5: Compare with domain adaptation methods.**
>
> To enable fairness, we compare TimeID with baselines, which are transformed to their domain adaptation version by finetuning in Table 1. The results show that TimeID outperforms baselines in most cases, demonstrating its effectiveness. Further, we have conducted additional experiments with more domain adaptation methods, i.e., ADvSKM [3], CoDATS [4], and SASA [5]. The results in Table https://bashify.io/i/DDSBty show that TimeID achieves SOTA performance.
>
> [3]. Adversarial Spectral Kernel Matching for Unsupervised Time Series Domain Adaptation, IJCAI 2021.
>
> [4]. Multi-source deep domain adaptation with weak supervision for time-series sensor data, KDD 2020.
>
> [5]. Time series domain adaptation via sparse associative structure alignment, AAAI 2021.
>
> **W6: The method is over-engineered.**
>
> Generally, TimeID consists of three main modules: invariant disentangled feature learning (IDFL), proxy denoising, and knowledge distillation. Other components (e.g., Fourier decomposition, LLM-based proxy generation, and gradient-level alignment) are not independent but included in these three main modules. Each component aims to address a specific challenge. For example, the IDFL module can alleviate distribution shift. Further, the ablation study in Figure 4 shows that these three components are all useful for effective SF-TSF.
>
> **W7: Unclear writing and expression.**
>
> We will carefully proofread the manuscript and make several modifications throughout the paper, including the following:
> 1. Improving the logical transition in Introduction and Methodology, e.g., challenge refinement.
> 2. Incorporating motivations for key modules in Methodology, e.g., proxy denoising, is proposed to correct the LLM prediction, which is used to guide the target model. Further, we propose gradient-level alignment to enforce gradient consistency across disentangled components.

---

> > ### Author Rebuttal · Reviewer_mHjN · 2026-04-03
> >
> > I appreciate the authors’ clarifications and additional experimental claims. The rebuttal improves the framing of the paper, especially by making the challenges more forecasting-specific and by adding comparisons to domain adaptation methods. However, my main concerns are still not fully resolved, so I am inclined to keep my original score.
> >
> > In particular, the rebuttal still does not convincingly explain why LLMs are necessary for source-free TSF, rather than simply being a helpful external knowledge source. The motivation remains fairly generic and does not fully justify this design choice.
> >
> > I also remain unconvinced by the justification for invariant feature learning. The paper still relies on a strong assumption that transferable invariant temporal features exist across domains despite potentially large shifts in trend and seasonality, and the rebuttal mainly provides prior references and visualization rather than a more direct justification.
> >
> > Finally, the proxy denoising mechanism is still not sufficiently justified to me. While the rebuttal clarifies that it is different from pseudo-labeling, it remains unclear why the discrepancy between source and target models should serve as a reliable estimate of LLM error. As a result, the overall method still feels somewhat over-composed relative to the clarity of its core innovation.
> >
> > Overall, while the rebuttal is helpful, it does not sufficiently address my main conceptual concerns regarding motivation, assumptions, and method justification, so I do not feel persuaded to change my original assessment.

---

> > > ### Author Response · Authors · 2026-04-05
> > >
> > > Thank you for the invaluable feedback. We would like to provide more clarifications as follows.
> > >
> > > * **Explain why LLMs are necessary for source-free TSF.**
> > >
> > > We would like to further clarify the necessity of LLMs for source-free time series forecasting (SF-TSF). In SF-TSF, without access to source data, the model lacks sufficient guidance for adaptation, which creates an information gap. In this case, the model may easily suffer from knowledge collapse due to the absence of source domain knowledge. Considering that time series and text share similar sequential characteristics, LLMs are widely adopted by existing works for time series forecasting [1-2]. We introduce an LLM-based TSF model as an external temporal knowledge source to mitigate the information gap, which is expected to provide complementary patterns that are not obvious in the limited target data. It benefits from powerful LLM's pretraining on massive corpora. In addition, we conduct an experiment using TimeID without LLM. As shown in Table 1, the results show that removing LLM leads to a consistent performance degradation, which demonstrates LLM's effectiveness for SF-TSF.
> > >
> > > [1]. Time-LLM: Time series forecasting by reprogramming large language models, ICLR 2024.
> > >
> > > [2]. One fits all: Power general time series analysis by pretrained lm, NeurIPS 2023.
> > >
> > > Table 1: Performance comparison of TimeID and its variant without LLM
> > >
> > > |Method|w/o LLM|TimeID|
> > > |---|---|---|
> > > |Metric|MSE/MAE|MSE/MAE|
> > > |ETTh1→ETTh2|0.292/0.350|0.280/0.338|
> > > |ETTh1→ETTm1|0.373/0.408|0.359/0.390|
> > > |ETTh1→ETTm2|0.187/0.276|0.177/0.267|
> > > |ETTh1→ECL|0.189/0.283|0.170/0.276|
> > > |ETTh1→Traffic|0.463/0.337|0.452/0.327|
> > > |ETTh1→Weather|0.180/0.239|0.169/0.228|
> > >
> > > * **Provide justification for invariant feature learning.**
> > >
> > > Regarding the justification for invariant disentangled feature learning (IDFL), we provide prior references and visualization during the rebuttal to verify the invariant disentangled feature learning from both theoretical and empirical perspectives. We would like to clarify that the IDFL does not assume the raw signals are invariant. It seeks invariance in the decomposed space. By disentangling trend and seasonality components, we capture component-invariant features while allowing component-specific cues to vary. Specifically, we capture the invariant trend component when seasonality changes or vice versa. In addition, we conduct an experiment using TimeID without IDFL. As shown in Table 2, the results show that removing IDFL leads to a consistent performance degradation, which demonstrates IDFL's effectiveness for SF-TSF.
> > >
> > > Table 2: Performance comparison of TimeID and its variant without IDFL
> > >
> > > |Method|w/o IDFL|TimeID|
> > > |---|---|---|
> > > |Metric|MSE/MAE|MSE/MAE|
> > > |ETTh1→ETTh2|0.311/0.372|0.280/0.338|
> > > |ETTh1→ETTm1|0.365/0.391|0.359/0.390|
> > > |ETTh1→ETTm2|0.189/0.275|0.177/0.267|
> > > |ETTh1→ECL|0.182/0.292|0.170/0.276|
> > > |ETTh1→Traffic|0.543/0.415|0.452/0.327|
> > > |ETTh1→Weather|0.195/0.254|0.169/0.228|
> > >
> > > * **The proxy denoising mechanism is still not sufficiently justified.**
> > >
> > > The key intuition behind the proxy denoising (PD) is that LLM forecasting errors in TSF are often systematic rather than purely random, such as amplitude or phase mismatch due to domain differences. The denoising step aims to correct these systematic biases by leveraging the discrepancy between source and target predictors, which provides a reference for how predictions should be adjusted under domain shift. In this sense, the procedure does not simply remove noise, but refines LLM outputs toward more consistent temporal dynamics. In addition, we conduct an experiment using TimeID without PD. As shown in Table 3, the results show that removing PD leads to a consistent performance degradation, which demonstrates PD's effectiveness for SF-TSF.
> > >
> > > Table 3: Performance comparison of TimeID and its variant without PD
> > >
> > > |Method|w/o PD|TimeID|
> > > |---|---|---|
> > > |Metric|MSE/MAE|MSE/MAE|
> > > |ETTh1→ETTh2|0.300/0.390|0.280/0.338|
> > > |ETTh1→ETTm1|0.453/0.438|0.359/0.390|
> > > |ETTh1→ETTm2|0.201/0.287|0.177/0.267|
> > > |ETTh1→ECL|0.201/0.299|0.170/0.276|
> > > |ETTh1→Traffic|0.470/0.340|0.452/0.327|
> > > |ETTh1→Weather|0.187/0.246|0.169/0.228|
> > >
> > > * **The overall method still feels over-composed.**
> > >
> > > TimeID is tailored for SF-TSF, which consists of three core modules, invariant disentangled feature learning (IDFL), proxy denoising, and knowledge distillation. Each component is designed to address a specific challenge in the source-free setting. First, to address distribution shift, we design the IDFL module. Then, to address the challenge of lacking source domain knowledge in SF setting, we need to employ LLMs to guide the target model, therefore we design a proxy denoising mechanism to improve the reliability of LLM-generated signals, and a knowledge distillation module to transfer refined information from LLMs to target model. These components are complementary rather than redundant. What's more, the ablation experiment results confirm the necessity of each module.

---

### Official Review · Reviewer_yfT4 · 2026-03-12

**Soundness:** 3
**Presentation:** 2
**Significance:** 3
**Originality:** 2
**Overall Recommendation:** 5
**Confidence:** 4

**Summary:**

This paper proposes TimeID for source-free time series forecasting, combining invariant representation learning and LLM-centric proxy denoising. TimeID includes dual-branch invariant-feature disentanglement based on season-trend decomposition, parameter-free proxy denoising to correct LLM prediction biases, and bidirectional KD alignment. Experiments on widely used time-series forecasting benchmarks demonstrate its effectiveness.

**Compliance With Llm Reviewing Policy:**

Affirmed.

**Final Justification:**

I appreciate the authors’ effort in preparing a clear and thorough rebuttal. The added explanations on the method, related discussions, and experiments have effectively addressed the concerns I previously raised. The response further enhances my confidence in quality and technical soundness of this paper. Thus, I would keep my positive attitude towards acceptance.

**Key Questions For Authors:**

Please refer to Weaknesses.

**Limitations:**

Yes

**Strengths And Weaknesses:**

## **Strengths**

* The authors introduces a new setting where adaption occurs without access to source data, which is make sense in real-world time series applications.

* TimeID includes three complementary techniques to learn dynamic time series, including invariant feature learning, proxy learning, and knowledge distillation.

* The authors conduct experiments to demonstrate TimeID's effectiveness and further explore detailed hyperparameter sensitivity.

## **Weaknesses**

* This paper lacks evidence showing how LLM prediction errors correlate with the forecasting biases. What is even more confusing is under what conditions this proxy denoising strategy can outperform standard denoising techniques. It would be better to compare the proposed proxy denoising with other denoising strategies.

* My primary concern lies with Eq. 18 (optimization objective), which defines an objective that contains 6 terms (one primary objective plus five constraints). Despite the authors providing extensive hyperparameter sensitivity analysis, this inevitably introduces complexity. This means TimeID is more vulnerable to optimization. From another perspective, this highly complex optimization objective undermines the credibility of the robust results obtained by TimeID. This should be taken seriously.

* TimeID is a stack of existing techniques, such as seasonal trend decomposition (the most commonly used strategy in TS), existing predictors, Fourier transform-based TS modeling, and gradient alignment. Although the domain adaptation problem is relatively novel, the experiments deliberately overstate this point. For instance, the transfers within the ETT dataset (such as ETTh1 to ETTh2) are meaningless because they exhibit fully transferable patterns.

* Following the above point, zero-shot adaptation should be more suitable for this problem formulation, but TimeID does not employ it. In addition, it would be better to consider more time-series datasets for forecasting.

---

> ### Author Rebuttal · Authors · 2026-03-31
>
> Thanks for the constructive comments.
>
> **W1: Compare with other denoising strategies.**
>
> LLM prediction errors are often structured and domain-dependent rather than purely stochastic noise [1, 2]. Although their errors often manifest as systematic biases (e.g., shifted amplitude or phase misalignment), these biases relate to domain differences in most cases. Thus, the prediction errors correlate with the forecasting biases due to the domain difference between the source and target datasets.
>
> We conduct additional experiments against other denoising techniques. As shown in Table https://bashify.io/i/GfAv9b, the results show that proxy denoising outperforms other methods. The standard denoising method focuses on removing high-frequency random noise but cannot correct semantic forecasting shifts caused by domain gaps. The proposed proxy denoising is a knowledge-level correction rather than a signal-level filter. Therefore, it outperforms standard methods when the noise is systematic distribution shifts.
>
> We will clarify the correlation between LLM prediction errors and forecasting biases in the Methodology section and add the results in the Experiments section.
>
> [1]. Large language models are zero-shot time series forecasters, NeurIPS 2023.
>
> [2]. LangTime: A Language-Guided Unified Model for Time Series Forecasting with Proximal Policy Optimization, ICML 2025.
>
> **W2: The optimization objective is overly complex.**
>
> While the final objective contains multiple terms, each component is designed to address a specific aspect of the source-free setting. These terms are not independent but work together to regularize the model from different perspectives. Regarding optimization stability, the sensitivity analysis results in the paper show that the method is relatively stable across a range of hyperparameters. We also find that a consistent set of hyperparameters works well across datasets without extensive tuning. The training loss curve (please see Figure https://bashify.io/i/BQwMZX) demonstrates that the optimization process is robust and stable. We will add the training loss curve in the Experiments section to confirm that the optimization is stable.
>
> **W3: Stack of existing techniques.**
>
> While TimeID integrates several established modules, the core innovation lies in proposing a unified framework tailored to the source-free time series forecasting (SF-TSF). Existing decomposition or alignment methods are typically used in supervised or standard DA settings where source data is available. Our contribution is the first to propose an LLM-centric Proxy-Denoise-Distill pipeline for SF-TSF. The innovation lies in introducing entirely new individual components, and in how these modules are re-engineered to facilitate knowledge transfer when source data is completely unavailable. Regarding the transfers within the ETT dataset, such as ETTh1 to ETTh2, we include such settings to align with standard protocols in the TSF community and represent a relatively mild domain shift. Furthermore, we also conduct more cross-dataset transfer experiments (e.g., transferring from Weather to Electricity, or Traffic to Weather) in Appendix A3.1, as shown in Table https://bashify.io/i/1PlR0E. These scenarios involve different temporal patterns and provide a richer evaluation of our model's adaptation capabilities. We will further emphasize the core innovation in Introducion section.
>
> **W4: Zero-shot adaptation and more datasets.**
>
> Zero-shot adaptation is indeed an interesting direction for this problem. However, the current focus is on a practical setting where a small amount of target data is available, which is common in real-world forecasting scenarios. A zero-shot experiment is conducted, as shown in Table https://bashify.io/i/73Mcp3, and the results indicate that TimeID does not perform as well as in the original setting. This is because the method is specifically designed for few-shot scenarios. The core components (e.g., proxy denoising and knowledge distillation) operate during the adaptation training process. In the zero-shot setting, these modules are not applicable, leading to a performance decline of TimeID.
>
> Additional experiments are conducted on the Exchange, Solar, and PEMS datasets, as shown in Table https://bashify.io/i/6CjGn9. The results show that TimeID achieves consistent performance improvements across all datasets, further demonstrating the effectiveness of the proposed method. These results will be included in the Experiments section.

---

> > ### Author Rebuttal · Reviewer_yfT4 · 2026-04-01
> >
> > I would like to thank the authors for their efforts during the rebuttal process. My concerns have been addressed.
> >
> >  I suggest that the author further analyse the visualisations provided during the rebuttal in the final version to strengthen the overall argument. Furthermore, it would be better to discuss in greater detail how the work adapts to scenarios involving TS foundation models.
> >
> > I have raised the score to recommend acceptance.

---

> > > ### Author Response · Authors · 2026-04-06
> > >
> > > Thank you for the positive feedback. We would like to provide more clarifications about the additional comments.
> > >
> > > * **Analyse the visualisations provided during the rebuttal.**
> > >
> > > We would like to further analyze the visualisations (see https://bashify.io/i/BQwMZX) provided during the rebuttal. The training loss curve illustrates the optimization process of TimeID on ETTh1 $\rightarrow$ ETTm1. We find that the curve drops steadily within the first 5 epochs and then converges. The entire process shows no significant fluctuations, which indicates that the optimization of TimeID is stable and robust.
> > >
> > > * **Discuss how the work adapts to scenarios involving TS foundation models.**
> > >
> > > We would like to discuss how the work adapts to scenarios involving TS foundation models as follows. The framework is designed to be model-agnostic and can be naturally extended to scenarios involving time series foundation models. In particular, the LLM-based proxy model does not rely on a specific type of predictor, and can be replaced by TS foundation models. We conduct additional experiments using the time series foundation model (e.g., Chronos [1]) to replace the LLM-based time series forecasting model. As shown in Table 1, we find that performance declines when the LLM is replaced with the time series foundation model in most cases. This may be because that TS foundation models (e.g., Chronos) are designed to capture general temporal patterns, but their capability to provide robust cross-domain transferable knowledge remains limited, which shows that LLMs are more suitable for source-free time series forecasting.
> > >
> > > [1] Chronos: Learning the Language of Time Series, TMLR 2024.
> > >
> > > Table 1: Performance comparison of TSFM-based proxy model and LLM-based proxy model
> > >
> > > |Method|TSFM-based proxy model|LLM-based proxy model (TimeID)|
> > > |---|---|---|
> > > |Metric|MSE/MAE|MSE/MAE|
> > > |ETTh1→ETTh2|0.301/0.354|0.280/0.338|
> > > |ETTh1→ETTm1|0.439/0.432|0.359/0.390|
> > > |ETTh1→ETTm2|0.181/0.272|0.177/0.267|
> > > |ETTh1→Weather|1.465/0.700|0.169/0.228|
> > > |ETTh1→Electricity|0.169/0.276|0.170/0.276|
> > > |ETTh1→Traffic|0.461/0.338|0.452/0.327|
> > >
> > > We will incorporate the above discussion in the final version.

---

### Official Review · Reviewer_wiij · 2026-03-24

**Soundness:** 3
**Presentation:** 4
**Significance:** 4
**Originality:** 4
**Overall Recommendation:** 5
**Confidence:** 4

**Summary:**

This paper addresses the problem of source-free time series forecasting, assuming that only a pretrained source model is accessible while the target domain only provides scarce data. The authors propose TimeID consisting of three innovative components. First, it learns invariant and disentangled representations by explicitly decomposing time series into trend and seasonal factors, while enforcing cross-domain consistency. Second, it introduces a proxy denoising mechanism that regards the LLM-based forecaster as a noisy supervisor and improves its predictions by leveraging the agreement between source and target models. Third, it applies knowledge distillation to transfer the refined, high-level temporal knowledge into a compact target model. Extensive experiments on cross-domain datasets demonstrate that TimeID consistently outperforms existing baselines.

**Compliance With Llm Reviewing Policy:**

Affirmed.

**Final Justification:**

After reading the rebuttal, I find that the authors have fully addressed my main concerns, and the clarifications help improve my understanding of the work. Overall, the rebuttal largely alleviate my concerns, and I therefore increase my score.

**Key Questions For Authors:**

1. Can you clarify whether all baselines are evaluated under strictly source-free conditions, and ensure that no source data is used during adaptation?
2. How sensitive is the method to the gradient masking threshold?
3. Have you evaluated whether proxy denoising remains effective when the source model exhibits strong domain bias?

**Limitations:**

yes

**Strengths And Weaknesses:**

1. The authors propose a novel source-free time series forecasting framework, which can transfer learned knowledge without access to source data. This framework provides an interesting direction to integrate time series forecasting and domain adapation, given that most existing domain adaptation methods directly rely on source data.
2. The proposed framework is comprehensive, encompassing decomposition-based invariant disentangled feature learning, proxy denoising and knowledge distillation. All these components are designed to address specific challenges in source-free adaptation and they are internally correlated.
3. The experimental evaluations show the consistent performance improvements over baselines across multiple datasets. This consistency suggest that the proposed method can generalize across different temporal settings, which strengthens the empirical validity of TimeID.
4. Efficiency analysis enhances the practical relevance of the work and ablation studies indicate that each component contributes meaningfully. This helps understand the trade-offs between performance and computational overhead and makes it easier to understand the role of each individual module.

1. The motivation for gradient-level invariance lacks deeper empirical analysis explaining why it is superior to representation-level alignment alone. Without further analysis, it is difficult to determine whether the performance improvement is benefited from the gradient-level invariance.
2. The proxy denoising mechanism assumes that the consensus between source and target predictors provides a reliable correction signal, but it would be better to discuss the scenario when the source and target predictors are biased.
3. The $\alpha$-percentile plays an important role in the gradient difference mask selection. It would be better to conduct more experiments to test the effect of $\alpha$-percentile.

---

> ### Author Rebuttal · Authors · 2026-03-31
>
> Thank you for the invaluable suggestions.
>
> **W1: Empirical justification to show that the combination of gradient-level invariance and representation-level alignment is better than representation-level alignment only.**
>
> We conduct additional ablation studies to test the effect of gradient-level invariance (GI). The results are shown as follows.
>
> Table 1. Performance comparison of TimeID and removing gradient invariance
> |Dataset|ETTh1→ETTh2|ETTh1→ETTm1|ETTh1→ETTm2|ETTh1→Weather |ETTh1→ECL|ETTh1→Traffic|
> |---|---|---|---|---|---|---|
> |Metric|MSE/MAE|MSE/MAE|MSE/MAE|MSE/MAE|MSE/MAE|MSE/MAE|
> |w/o_gradient invariant|0.281/0.340|0.397/0.413|0.184/0.274|0.174/0.231|0.175/0.284|0.460/0.345|
> |TimeID|0.280/0.338|0.359/0.390|0.177/0.267|0.169/0.228|0.170/0.276|0.452/0.327|
>
> The experimental results show that removing GI leads to consistent performance drops across datasets, demonstrating its effectiveness. TimeID employs representation-level alignment to learn similar representations across different components, but this could not constrain the training dynamics. The gradient-level invariance explicitly enforces consistent gradient learning across components, which allows the model to update parameters according to component-invariant patterns.
>
> **W2: The reliability of proxy denoising under biased source and target predictors is not discussed.**
>
> The proxy denoising strategy relies on the consensus between the source predictor and target predictor, which reduces the influence of individual model bias. The correction strength $\alpha$ is set to 1 when the consensus between the source and target predictors is fully trusted. When the source and target predictors are biased, the correction strength $\alpha$ can be reduced. Figure 17 (https://bashify.io/i/rNKYyF) in Appendix 3.11 also illustrates the effect of varying the correction strength $\alpha$ from 0 to 1. As the correction strength decreases, performance declines to some extent, yet remains within an acceptable range. This indicates that the proposed method remains effective even when both source and target predictors are biased.
>
> **W3: The effect of the $\alpha$-percentile in gradient mask selection is insufficiently explored.**
>
> The $\alpha$-percentile controls the sparsity of the gradient difference mask used for selecting reliable gradients, and an additional sensitivity analysis is conducted on it. As shown in Table 2, beyond the extreme case of 0.1, the method shows high stability and strong performance across the range of 0.3 to 0.9, indicating that the approach is not overly sensitive to this hyperparameter. We observe that higher drop rates, such as 0.7 or 0.9, yield the best results for datasets like Traffic and ETTh2, suggesting that focusing on a core subset of highly invariant features is often beneficial. We will add these results into Appendix to further analyze the effect of $\alpha$-percentile.
>
> Table 2. Effect of $\alpha$-percentile.
> |$\alpha$-percentil|0.1|0.3|0.5|0.7|0.9|
> |---|---|---|---|---|---|
> |Metric|MSE/MAE|MSE/MAE|MSE/MAE|MSE/MAE|MSE/MAE|
> |ETTh1→ETTh2|0.283/0.346|0.280/0.338|0.280/0.338|0.278/0.336|0.275/0.334|
> |ETTh1→ETTm1|1.676/0.910|0.359/0.390|0.361/0.390|0.385/0.404|0.375/0.394|
> |ETTh1→ETTm2|0.251/0.338|0.177/0.267|0.177/0.265|0.176/0.262|0.180/0.263|
> |ETTh1→ECL|0.175/0.283|0.170/0.276|0.165/0.270|0.156/0.259|0.155/0.256|
> |ETTh1→Traffic|0.468/0.343|0.452/0.327|0.448/0.330|0.443/0.323|0.440/0.321|
> |ETTh1→Weather|0.182/0.244|0.169/0.228|0.170/0.227|0.176/0.229|0.164/0.216|
>
> **Q1: It is unclear whether all baselines strictly follow the source-free setting without using source data during adaptation.**
>
> All baselines are evaluated under the strict source-free setting, where only the pretrained source model is available, and no source data is used during adaptation. The target domain data are used in the same manner as in our method, ensuring a fair comparison. We will clarify the settings in the Experiments section.
>
> **Q2: How sensitive is the method to the gradient masking threshold?**
>
> The gradient masking threshold is determined by the α-percentile, which controls how many gradients are selected as reliable signals. As shown in Table 2, we observe that performance remains stable across a reasonable range of thresholds. This suggests that the proposed mechanism is not highly sensitive to the exact threshold value.
>
> **Q3: Have you evaluated whether proxy denoising remains effective when the source model exhibits strong domain bias?**
>
> As discussed in W2, the correction strength $\alpha$ can be reduced when the source model exhibits strong domain bias. The results in Figure 17 in the Appendix show that as the correction strength decreased, the performance declined to some extent, but remained within an acceptable range. It indicates that the proxy denoising remains effective when the source model exhibits strong domain bias. We will clarify this in the Methodology section.

---

> > ### Author Rebuttal · Reviewer_wiij · 2026-04-01
> >
> > I would like to thank the authors for their diligent efforts of rebuttal. I am satisfied with the comprehensive additional experimental results and discussions. The rebuttal has addressed my problems. I decide to increase my socres. Please remember to incorporate the discussions into the final version.

---

> > > ### Author Response · Authors · 2026-04-01
> > >
> > > Dear Reviewer wiij,
> > >
> > > Thanks again for your insightful comments and positive feedback on our rebuttal. We promise to add the new experimental results and more discussions in the final version following your comments.
> > >
> > > Sincerely,
> > >
> > > Authors

---

### Decision · Program_Chairs · 2026-04-30

**Decision:**

Accept (regular)

**Comment:**

The paper proposes TimeID, a source-free time series forecasting framework that combines invariant disentangled representation learning, LLM-centric proxy denoising, and knowledge distillation to enable cross-domain adaptation without access to source data.

The paper addresses a significant research problem and proposes a well-motivated framework that integrates multiple complementary components. Reviewers found the approach to be technically sound, with a coherent design where each module targets a specific challenge in the source-free setting. The empirical evaluation is comprehensive, demonstrating consistent improvements over strong baselines across multiple datasets and architectures. The authors provided thorough and convincing rebuttal responses, adding additional experiments and clarifications that resolved most reviewer concerns, particularly regarding component effectiveness, robustness, and validity.

While some concerns remain regarding the complexity of the framework, assumptions behind invariant representation learning, and the conceptual justification of certain components, these issues are outweighed by the overall technical quality and empirical strength. The majority of reviewers converged to clearly positive recommendations after the rebuttal, indicating a strong consensus in favor of acceptance. Therefore, the paper is accepted.